# Improving Generative Adversarial Imitation Learning with Non-expert Demonstrations

## Abstract

Imitation learning aims to learn an optimal policy from expert demonstrations and its recent combination with deep learning has shown impressive performance. However, collecting a large number of expert demonstrations for deep learning is time-consuming and requires much expert effort. In this paper, we propose a method to improve generative adversarial imitation learning by using additional information from non-expert demonstrations which are easier to obtain. The key idea of our method is to perform multiclass classification to learn discriminator functions where non-expert demonstrations are regarded as being drawn from an extra class. Experiments in continuous control tasks demonstrate that our method learns better policies than the generative adversarial imitation learning baseline when the number of expert demonstrations is small.

## 1 Introduction

The goal of sequential decision making problems is to learn an optimal policy that exhibits task-solving behavior. Reinforcement learning (RL) is a powerful approach to find such a policy by maximizing rewards computed by a reward function (Puterman, 1994; Sutton & Barto, 1998). While RL has achieved great success in solving challenging tasks (Mnih et al., 2015; Silver et al., 2017), its performance depends heavily on a *good* reward function which well captures the concept of task-solving behavior. Unfortunately, designing such a good reward function is a difficult trial-and-error process and often time-consuming. This difficulty is one of the major limitations of RL for many real-world applications.

Imitation learning (IL) (Schaal, 1999) is an alternative approach to learn an optimal policy. In contrast to RL, IL has access to expert demonstrations, which are task-solving trajectories collected from experts who have mastered the task, and IL finds a policy that generates trajectories similar to these expert demonstrations. IL has been a long-studied problem and is attracting more attention recently, particularly in robotics (Duan et al., 2017; Stadie et al., 2017) and games (Ross et al., 2011). However, traditional IL methods rely on extensive feature engineering which makes their applicability quite limited.

Many approaches were proposed to overcome the difficulty of feature engineering in IL. Among them, the most successful approach is to use deep neural networks to learn representative features in an end-to-end manner (Wulfmeier et al., 2015; Finn et al., 2016a; Ho & Ermon, 2016; Fu et al., 2018). In particular, *generative adversarial imitation learning* (GAIL) (Ho & Ermon, 2016) is a state-of-the-art method that uses generative adversarial training to perform IL with deep neural networks. While the combination of IL and deep learning led to impressive performance improvement, it has introduced a new limitation regarding sample efficiency as training a deep neural network generally requires a large amount of data. This is a severe limitation in IL since collecting a large number of expert demonstrations can be expensive and time-consuming, and requires much expert effort. Moreover, the generative adversarial training procedure is known to be highly unstable (Mescheder et al., 2018), and this issue becomes more severe when only a small amount of data is available.

While expert demonstrations may be expensive, *non-expert* demonstrations collected from non-experts who have not mastered the task are often much cheaper to obtain. For instance, demonstrations from amateur-level players in the game of Go are much cheaper to obtain than those from master-level players. Leveraging additional information from a large number of non-expert demonstrations to improve IL is the key idea of semi-supervised inverse RL (SSIRL) (Valko et al., 2012)

and IRL from failure (IRLF) (Shiarlis et al., 2016). Both SSIRL and IRLF have shown to perform well for low-dimensional problems with proper feature engineering. However, they are extensions of traditional IL methods and are not capable of efficiently training deep neural networks which are needed for handling high-dimensional problems. Moreover, both of them rely on rather restrictive assumptions about data generating processes of non-expert demonstrations which we cannot control in practice.

In this paper, we propose a novel method to leverage non-expert demonstrations without the afore-mentioned weaknesses of SSIRL and IRLF. Our method is built upon the generative adversarial training procedure where we perform multiclass classification to learn discriminator functions with non-expert demonstrations regarded as being drawn from an extra class. Our method uses both expert and non-expert demonstrations in the discriminator learning objective, and this leads to a better feature representation of discriminator functions. We show that the minimax formulation commonly used in generative adversarial training does not guarantee the optimality of policies for our method, and we alternatively propose a modified optimization procedure which provides such a guarantee. We also show that naive extensions of GAIL that mix non-expert demonstrations with expert demonstrations or agent's trajectories only learn a mixture policy and does not learn the expert policy. Experiments on benchmark continuous control tasks show that our method performs better than GAIL especially when only a small number of expert demonstrations is available.

## 2 RELATED WORK

IL has been a long-studied problem and there are many approaches to solve this problem, including behavior cloning (Pomerleau, 1988), occupancy measure matching (Syed et al., 2008) and IRL (Russell, 1998; Ng & Russell, 2000). Recently, IL methods that use a generative adversarial training procedure (Goodfellow et al., 2016) have gained a great deal of interest thanks to its effectiveness at training deep neural networks (Ho & Ermon, 2016; Finn et al., 2016a; Fu et al., 2018). Despite such success, deep neural networks are well-known to have poor data efficiency and require a large amount of data to train. For this reason, these methods may not perform well when only a small number of expert demonstrations are available.

Semi-supervised learning (SSL) (Chapelle et al., 2010) improves sample efficiency by utilizing a large amount of unlabeled data and it has shown promising results in deep learning (Ranzato & Szummer, 2008; Weston et al., 2012). While mainly developed for supervised learning, SSL can be applied to improve some IL methods as well. In particular, *semi-supervised IRL* (SSIRL) (Valko et al., 2012) improves the IRL method of Abbeel & Ng (2004) by using semi-supervised support vector machines to classify between expert demonstrations and trajectories generated by the agent. However, this approach is not suitable due to the difference in data generating processes between SSL and IRL. More specifically, SSL methods generally assume that an unlabeled dataset is a mixture of positive and negative samples. For SSIRL, this implies that an unlabeled demonstration dataset is a mixture of expert demonstrations and the agent's trajectories. However, collecting such an unlabeled dataset is quite difficult in practice since we do not know the agent's policies before-hand. This issue has been remedied to some extent by Audiffren et al. (2015) where the authors proposed using a manifold regularization technique which relies on a milder assumption on the unlabeled dataset. However, manifold regularization requires an appropriate similarity function to perform well. Moreover, both methods are unsuitable in high-dimensional problems due to its dependence on the linearity of reward functions and good feature engineering.

*IRL from failure* (IRLF) (Shiarlis et al., 2016) also utilizes an additional demonstration dataset which is assumed to consist of demonstrations collected by non-expert who *failed* to solve the task. Using this dataset, the authors proposed an IRL method that encourages the agent to be dissimilar to non-expert while learning an expert policy. While IRLF is technically sounded, collecting strictly failure demonstrations can be expensive on tasks such as autonomous driving where failures are catastrophic. Moreover, this method is still restricted by the linear reward assumption and is not applicable to train deep neural networks.

Using additional datasets to improve generative adversarial training was explored in the context of semi-supervised generative modeling (Salimans et al., 2016; Li et al., 2017a) and multi-modal generative modeling (Liu & Tuzel, 2016). However, if we want to apply these methods to our IL setting, then we are required to generate trajectories to imitate non-experts. This is inefficient since

generating trajectories requires interactions with environment and we would like to keep the number of such interactions as small as possible. In contrast, our method only learns the expert policy and only generate trajectories to imitate the expert.

Our proposal of using an additional dataset as an extra class in multiclass classification resembles the idea of *universum learning* (Vapnik, 1998; 2006; Zhang & LeCun, 2017). So far, universum learning has been applied only to supervised learning problems, especially for discriminative learning with support vector machines. Thus, our contribution may be regarded as the first attempt to apply the idea of universum learning to IL and also to generative adversarial learning.

## 3 BACKGROUND

In this section, we provide backgrounds of RL, IL, and GAIL.

### 3.1 REINFORCEMENT LEARNING (RL)

An RL problem is formulated as a discrete-time Markov decision process (MDP) which is defined by a tuple $\mathcal{M} = (\mathcal{S}, \mathcal{A}, p(\mathbf{s}'|\mathbf{s}, \mathbf{a}), p_0(\mathbf{s}), r(\mathbf{s}, \mathbf{a}), \gamma)$, where $\mathcal{S} \subseteq \mathbb{R}^{d_\mathbf{s}}$ is the (continuous) state space, $\mathbf{s} \in \mathcal{S}$ is a state, $\mathcal{A} \subseteq \mathbb{R}^{d_\mathbf{a}}$ is the (continuous) action space, $\mathbf{a} \in \mathcal{A}$ is an action, $p(\mathbf{s}'|\mathbf{s}, \mathbf{a})$ is the transition probability density from $\mathbf{s}$ to $\mathbf{s}'$ when $\mathbf{a}$ is taken, $p_0(\mathbf{s})$ is the initial state probability density, $r(\mathbf{s}, \mathbf{a})$ is the reward function, and $0 < \gamma \leqslant 1$ is the discount factor[1]. In each discrete time-step $t \geqslant 0$, an agent in a state $\mathbf{s}_t$ chooses an action $\mathbf{a}_t$ according to a policy $\pi(\mathbf{a}|\mathbf{s}_t)$, which is a conditional probability density. Then, the agent transits to a next state $\mathbf{s}_{t+1} \sim p(\mathbf{s}'|\mathbf{s}_t, \mathbf{a}_t)$ and receives a reward $r(\mathbf{s}_t, \mathbf{a}_t)$. We call a sequence of states and actions a trajectory $\boldsymbol{\tau}$. The goal of RL is to find an optimal policy that maximizes the expected discounted cumulative rewards (also called return) defined as

$$\mathbb{E}_{p_0(\mathbf{s}_0), \pi(\mathbf{a}_t|\mathbf{s}_t)_{t \geqslant 0}, p(\mathbf{s}_{t+1}|\mathbf{s}_t, \mathbf{a}_t)_{t \geqslant 0}} \left[ \sum_{t=0}^{\infty} \gamma^t r(\mathbf{s}_t, \mathbf{a}_t) \right] = \mathbb{E}_\pi \left[ r(\mathbf{s}, \mathbf{a}) \right], \tag{1}$$

where the expectation is taken over the probability densities for all time steps and we use the notation $\mathbb{E}_\pi$ for brevity. When the policy is a parameterized function with parameter $\boldsymbol{\theta}$, a locally optimal policy can be found by optimization methods such as policy gradients (Williams, 1992).

We also use an equivalent formulation in terms of *occupancy measures* (Puterman, 1994; Altman, 1999; Syed et al., 2008). A state-action occupancy measure defines the expected discounted (unnormalized) visitation density of each state-action pair and is denoted by $\rho_\pi(\mathbf{s}, \mathbf{a}) = \mathbb{E}_{p_0(\mathbf{s}_0), \pi(\mathbf{a}_t|\mathbf{s}_t)_{t \geqslant 0}, p(\mathbf{s}_{t+1}|\mathbf{s}_t, \mathbf{a}_t)_{t \geqslant 0}} [\sum_{t=0}^{T} \gamma^t \delta(\mathbf{s}_t - \mathbf{s}, \mathbf{a}_t - \mathbf{a})]$, where $\delta$ is the Dirac delta function[2]. An important property of the occupancy measure is that if it satisfies the Bellman flow constraints, $\int \rho_\pi(\mathbf{s}', \mathbf{a}') d\mathbf{a}' = p_0(\mathbf{s}') + \gamma \iint p(\mathbf{s}'|\mathbf{s}, \mathbf{a}) \rho_\pi(\mathbf{s}, \mathbf{a}) d\mathbf{s} d\mathbf{a}$, then there is one-to-one correspondence between the occupancy measure and a policy given as $\pi(\mathbf{a}|\mathbf{s}) = \rho_\pi(\mathbf{s}, \mathbf{a})/\rho_\pi(\mathbf{s})$, where $\rho_\pi(\mathbf{s}) = \int \rho_\pi(\mathbf{s}, \mathbf{a}) d\mathbf{a}$ is a state occupancy measure. This property allows us to rewrite the RL objective to be maximized as $\mathbb{E}_\pi \left[ r(\mathbf{s}, \mathbf{a}) \right] = \iint \rho_\pi(\mathbf{s}, \mathbf{a}) r(\mathbf{s}, \mathbf{a}) d\mathbf{s} d\mathbf{a}$.

The optimal policy of the above MDP is deterministic (Puterman, 1994) and a stochastic policy should be reduced to a deterministic policy at an optimum. However, a deterministic policy suffers from an exploration issue. In many tasks, it is beneficial to consider maximum entropy RL (Ziebart et al., 2008; 2010) whose optimal policy is a stochastic policy maximizing

$$\mathbb{E}_\pi \left[ r(\mathbf{s}, \mathbf{a}) \right] + \beta \mathcal{H}(\pi), \tag{2}$$

where $\beta \geqslant 0$ and $\mathcal{H}(\pi) = -\mathbb{E}_\pi \left[ \log \pi(\mathbf{a}|\mathbf{s}) \right]$ is a discounted causal entropy (Ziebart et al., 2010). The advantage of maximum entropy RL is that it encourages exploration and allows the agent to find a better policy when compared to the standard RL formulation (Haarnoja et al., 2017).

### 3.2 IMITATION LEARNING (IL)

The goal of IL, or apprenticeship learning, is to learn a parameterized policy $\pi_{\boldsymbol{\theta}}$, with policy parameter $\boldsymbol{\theta}$, such that $\pi_{\boldsymbol{\theta}}$ exhibits the same behavior as an expert policy $\pi_{\mathrm{E}}$. We assume that $\pi_{\mathrm{E}}$ is unknown.

---

[1]$\gamma = 1$ is only allowed for finite horizon setting.

[2]For discrete $\mathcal{S}$ and $\mathcal{A}$, $\delta$ is replaced by the indicator function.

We instead have access to expert demonstrations $\mathcal{D}_E = \{(\mathbf{s}_i, \mathbf{a}_i)\}_{i=1}^N$ which are trajectories generated by executing the expert policy under an MDP. IL methods can be categorized into interactive methods and non-interactive methods. Interactive methods such as structured prediction (Ross et al., 2011) allow the agent to query for expert demonstrations during learning. Despite their strong theoretical guarantees and great empirical performances, these methods require the expert to be available during learning which is not always possible in reality. On the other hand, non-interactive methods, such as occupancy measure matching (Syed et al., 2008) and IRL (Ng & Russell, 2000; Abbeel & Ng, 2004), only require a pre-collected demonstration dataset for learning. In this paper, we focus on the non-interactive IL setting due to its high practicality.

### 3.3 GENERATIVE ADVERSARIAL IMITATION LEARNING (GAIL)

GAIL (Ho & Ermon, 2016) is a state-of-the-art non-interactive IL method that performs occupancy measure matching to learn the parameterized policy. In occupancy measure matching (Syed et al., 2008), the policy parameter is learned to minimize a distance measure $\ell$ between occupancy measures of $\pi_E$ and $\pi_\theta$. More formally, occupancy measure matching methods solve an optimization problem $\min_\theta \ell(\rho_{\pi_E}, \rho_{\pi_\theta}) - \beta \mathcal{H}(\pi_\theta)$, where $\mathcal{H}$ is the causal entropy regularizer with $\beta \geqslant 0$.

The key idea of GAIL is to use generative adversarial training to estimate the distance and to minimize the estimated distance. Briefly speaking, the distance measure is the Jensen-Shannon (JS) divergence defined as $\mathrm{JS}(\rho_{\pi_E}, \rho_{\pi_\theta}) = \frac{1}{2}\left(\mathrm{gKL}(\rho_{\pi_E}||(\rho_{\pi_E} + \rho_{\pi_\theta})/2) + \mathrm{gKL}(\rho_{\pi_\theta}||(\rho_{\pi_E} + \rho_{\pi_\theta})/2)\right)$, where $\mathrm{gKL}(\rho||q) = \iint \rho(\mathbf{s}, \mathbf{a}) \log \frac{\rho(\mathbf{s},\mathbf{a})}{q(\mathbf{s},\mathbf{a})} \mathrm{ds}\mathrm{da} - \iint \rho(\mathbf{s}, \mathbf{a})\mathrm{ds}\mathrm{da} + \iint q(\mathbf{s}, \mathbf{a})\mathrm{ds}\mathrm{da}$ is the generalized Kullback-Leibler (gKL) divergence defined for unnormalized densities[3] (Dikmen et al., 2015). Since both occupancy measures are unknown, the divergence is approximated via a binary classification problem: $\max_\phi \mathbb{E}_{\pi_\theta}[\log D_\phi(\mathbf{s}, \mathbf{a})] + \mathbb{E}_{\pi_E}[\log(1 - D_\phi(\mathbf{s}, \mathbf{a}))]$. The function $D_\phi : \mathcal{S} \times \mathcal{A} \to (0, 1)$ is called a discriminator and is often parameterized by a deep neural network as $D_\phi(\mathbf{s}, \mathbf{a}) = \exp(d_\phi(\mathbf{s}, \mathbf{a}))/(\exp(d_\phi(\mathbf{s}, \mathbf{a})) + 1)$ where $d_\phi$ is an output of a deep neural network. It can be shown that if the discriminator has infinite capacity, the global maximum of this binary classification problem corresponds to the JS divergence up to a constant. Based on this fact, GAIL minimizes an approximated JS divergence by solving the following minimax optimization problem:

$$\min_\theta \max_\phi \mathbb{E}_{\pi_\theta}\left[\log D_\phi(\mathbf{s}, \mathbf{a})\right] + \mathbb{E}_{\pi_E}\left[\log(1 - D_\phi(\mathbf{s}, \mathbf{a}))\right] - \beta\mathcal{H}(\pi_\theta). \tag{3}$$

In practice, this optimization problem is solved by alternately performing gradient ascent for $\phi$ and gradient descent for $\theta$. Notice that the gradient descent step for $\theta$ is equivalent to performing policy gradient ascent with reward function $r(\mathbf{s}, \mathbf{a}) = -\log(D_\phi(\mathbf{s}, \mathbf{a}))$ and a causal entropy.

## 4 IMITATION LEARNING WITH NON-EXPERT DEMONSTRATIONS

GAIL is an effective method that achieves state-of-the-art performance for high-dimensional IL problems. However, the generative adversarial training procedure described above is known to be highly unstable (Mescheder et al., 2018). This instability issue becomes more apparent when only a small amount of data is available. One of the reasons is due to inaccuracy of estimating the divergence via a discriminator learned by using a small number of demonstrations. A classical approach to improve binary classification is to use an unlabeled dataset in the context of SSL. However, as discussed previously, SSL is not suitable for IL due to its assumption of unlabeled data.

Note that the discriminator in GAIL is learned using two sets of data samples; expert demonstrations and trajectories collected by agent. For this reason, even when the number of expert demonstrations is small, GAIL may still learn well after observing a sufficiently large number of agent's trajectories. However, this is not desirable since in practice we often prefer data efficient methods that can learn well even with a small number of agent's trajectories.

To improve discriminator learning with a milder assumption on an additional demonstration dataset, we propose to perform multiclass classification using the additional dataset as a new class. In the following, we first formally describe our problem setting, and then present our IL method that learns a multiclass classifier for generative adversarial training.

---

[3]An occupancy measure is not a density since it is not summed to one. Therefore, gKL is used instead.

### 4.1 Problem Setting

We consider an IL problem with an additional non-expert demonstration dataset as follows. We assume that expert demonstrations $\mathcal{D}_{\mathrm{E}} = \{(\mathbf{s}_i, \mathbf{a}_i)\}_{i=1}^N$ are trajectories collected by executing expert policy $\pi_{\mathrm{E}}$ in an MDP with reward function $r_{\mathrm{E}}(\mathbf{s}, \mathbf{a})$. We assume that $\pi_{\mathrm{E}}$ is an optimal policy, i.e., $\pi_{\mathrm{E}} = \mathrm{argmax}_\pi \mathbb{E}_\pi [r_{\mathrm{E}}(\mathbf{s}, \mathbf{a})]$. In addition, we also have non-expert demonstrations $\mathcal{D}_{\mathrm{N}} = \{(\mathbf{s}_j, \mathbf{a}_j)\}_{j=1}^M$ collected by executing non-expert policy $\pi_{\mathrm{N}}$ in the same MDP. The non-expert is assumed to be sub-optimal and gives worse expected rewards than the expert policy, i.e., $\mathbb{E}_{\pi_{\mathrm{N}}} [r_{\mathrm{E}}(\mathbf{s}, \mathbf{a})] < \mathbb{E}_{\pi_{\mathrm{E}}} [r_{\mathrm{E}}(\mathbf{s}, \mathbf{a})]$. We assume that $\mathcal{D}_{\mathrm{N}}$ is easier to obtain than $\mathcal{D}_{\mathrm{E}}$ and thus $M \gg N$. Our goal is to learn parameterized policy $\pi_{\boldsymbol{\theta}}$ that generates trajectories similar to those from $\pi_{\mathrm{E}}$ using both $\mathcal{D}_{\mathrm{E}}$ and $\mathcal{D}_{\mathrm{N}}$.

The main challenge for solving this problem is to efficiently leverage information brought by non-expert demonstrations to learn the expert policy. Under a very weak assumption that $\mathbb{E}_{\pi_{\mathrm{N}}} [r_{\mathrm{E}}(\mathbf{s}, \mathbf{a})] < \mathbb{E}_{\pi_{\mathrm{E}}} [r_{\mathrm{E}}(\mathbf{s}, \mathbf{a})]$, the non-expert policy can be totally unrelated to the expert policy and contains little to none of useful information that can improve learning. In the worst case, the non-expert policy could just be a random policy that randomly generates trajectories.

Therefore, to make efficient learning possible, we need a stronger but not too restrictive assumption on the non-expert demonstrations. Recall that both expert and non-expert demonstrations are collected under the same MDP, and this makes it intuitive to assume that state-action pairs of both the expert and non-expert lie on similar low-dimensional manifolds. To make this statement more explicit, we assume that there exists a feature map $\psi : \mathcal{S} \times \mathcal{A} \mapsto \mathbb{R}^b$ such that state-action pairs from the expert and non-expert are linearly separable in this feature space. While this assumption itself still does not allow us to always escape from the random policy case, it allows us to utilize reasonably generated non-expert demonstrations to improve representation learning when training a discriminator. Furthermore, this assumption also naturally implies that the quality of learned features is improved as the non-expert policy becomes more related to the expert policy since their low-dimensional manifolds becomes more similar to each other. This is the opposite to IRLF where the failure demonstrations should make performance as worse as possible.

### 4.2 Multiclass Classification for Discriminator Learning

Our generative adversarial method alternates between the discriminator learning step and policy learning step. In the discriminator learning step, our goal is to learn a multiclass probabilistic classifier that classifies state-action pairs into three classes; the expert class with label $y = \mathrm{E}$, the non-expert class with label $y = \mathrm{N}$, and the agent class with label $y = \mathrm{A}$. Let the following softmax models be estimates of the class posterior of the three classes:

$$F_\phi(\mathbf{s}, \mathbf{a}) = \frac{\exp(f_\phi(\mathbf{s}, \mathbf{a}))}{Z_\phi(\mathbf{s}, \mathbf{a})}, \quad G_\phi(\mathbf{s}, \mathbf{a}) = \frac{\lambda \exp(g_\phi(\mathbf{s}, \mathbf{a}))}{Z_\phi(\mathbf{s}, \mathbf{a})}, \quad H_\phi(\mathbf{s}, \mathbf{a}) = \frac{\exp(h_\phi(\mathbf{s}, \mathbf{a}))}{Z_\phi(\mathbf{s}, \mathbf{a})}, \quad (4)$$

where $F_\phi(\mathbf{s}, \mathbf{a})$ estimates the expert class posterior $p(y{=}\mathrm{E}|\mathbf{s}, \mathbf{a})$, $G_\phi(\mathbf{s}, \mathbf{a})$ estimates the non-expert class posterior $p(y{=}\mathrm{N}|\mathbf{s}, \mathbf{a})$, $H_\phi(\mathbf{s}, \mathbf{a})$ estimates the agent class posterior $p(y{=}\mathrm{A}|\mathbf{s}, \mathbf{a})$, and $Z_\phi(\mathbf{s}, \mathbf{a}) = \exp(f_\phi(\mathbf{s}, \mathbf{a})) + \lambda \exp(g_\phi(\mathbf{s}, \mathbf{a})) + \exp(h_\phi(\mathbf{s}, \mathbf{a}))$ is the normalization factor. $\phi$ is the parameter to be learned and $\lambda > 0$ is a weight parameter specified by the user to control the influence of non-expert demonstrations. To learn parameter $\phi$ for given policy parameter $\boldsymbol{\theta}$, we maximize the following objective

$$\mathcal{L}_\lambda(\boldsymbol{\phi}, \boldsymbol{\theta}) = \mathbb{E}_{\pi_{\mathrm{E}}} [\log F_\phi(\mathbf{s}, \mathbf{a})] + \lambda \mathbb{E}_{\pi_{\mathrm{N}}} [\log G_\phi(\mathbf{s}, \mathbf{a})] + \mathbb{E}_{\pi_{\boldsymbol{\theta}}} [\log H_\phi(\mathbf{s}, \mathbf{a})]. \quad (5)$$

Notice that the same $\lambda$ is used for both $G_\phi(\mathbf{s}, \mathbf{a})$ and $\mathcal{L}_\lambda(\boldsymbol{\phi}, \boldsymbol{\theta})$, and this choice is particularly important for our analyses in the following sections. We call the functions $f_\phi$, $g_\phi$, and $h_\phi$ discriminators. For smooth and differentiable discriminators, this objective can be maximized by stochastic gradient ascent or its adaptive step-size variants. The expectations over $\pi_{\mathrm{E}}$ and $\pi_{\mathrm{N}}$ are approximated using mini-batch samples from $\mathcal{D}_{\mathrm{E}}$ and $\mathcal{D}_{\mathrm{N}}$, respectively, while the expectation over $\pi_{\boldsymbol{\theta}}$ is approximated using trajectories $\{(\mathbf{s}_k, \mathbf{a}_k)\}_{k=1}^K$ collected by executing policy $\pi_{\boldsymbol{\theta}}$.

We explicitly assume that the discriminators are represented by deep neural networks with shared hidden layers and three outputs where each output represents $f_\phi$, $g_\phi$, and $h_\phi$. This parameterization allows us to explicitly make use of our assumption of using common feature map $\psi$. That is, the shared hidden layers learns a feature map which allows the three classes to be linearly separable, or

closed to be linearly separable. A large number of non-expert demonstrations allow us to accurately learn such a feature map which leads to more reliable discriminators and a better classification accuracy, as experimentally demonstrated in Section 6.

### 4.3 Occupancy Measure Matching with Multiclass Classifier

Our remaining task is to learn the parametrized policy in the policy learning step. Here, we present our occupancy measure matching method where a divergence is approximated from discriminators. First, we show that the discriminator learning objective approximates a JS divergence among three occupancy measures. This actually also implies that a commonly used minimax formulation is unsuitable since we do not recover the expert policy even in an infinite capacity setting. To cope with this problem, we propose a modified objective which allows us to recover the expert policy.

Let $\rho_{\pi_E}(\mathbf{s}, \mathbf{a})$, $\rho_{\pi_N}(\mathbf{s}, \mathbf{a})$, and $\rho_{\pi_\theta}(\mathbf{s}, \mathbf{a})$ be occupancy measures of the expert, non-expert, and agent, respectively. A JS divergence among the three occupancy measures with corresponding weights $\left(\frac{1}{2+\lambda}, \frac{\lambda}{2+\lambda}, \frac{1}{2+\lambda}\right)$ is defined as

$$\mathrm{JS}_\lambda(\rho_{\pi_E}, \rho_{\pi_N}, \rho_{\pi_\theta}) = \frac{1}{2+\lambda}\left(\mathrm{gKL}(\rho_{\pi_E}||\bar{q}) + \lambda \mathrm{gKL}(\rho_{\pi_N}||\bar{q}) + \mathrm{gKL}(\rho_{\pi_\theta}||\bar{q})\right), \quad (6)$$

where $\bar{q}(\mathbf{s}, \mathbf{a}) = (\rho_{\pi_E}(\mathbf{s}, \mathbf{a}) + \lambda\rho_{\pi_N}(\mathbf{s}, \mathbf{a}) + \rho_{\pi_\theta}(\mathbf{s}, \mathbf{a}))/(2 + \lambda)$ and $\mathrm{gKL}(\rho||\bar{q})$ is the generalized KL divergence. Note that Eq.(6) may be defined by either KL or gKL since the extra terms in gKL cancel out. Then, under the assumption that the discriminators have infinite capacity, the optimal parameter $\phi^\star$ which maximizes $\mathcal{L}_\lambda(\phi, \theta)$ satisfies

$$\mathcal{L}_\lambda(\phi^\star, \theta) = (2 + \lambda)\mathrm{JS}_\lambda(\rho_{\pi_E}, \rho_{\pi_N}, \rho_{\pi_\theta}) + \log\left(\lambda^\lambda(2 + \lambda)^{-(2+\lambda)}\right). \quad (7)$$

The proof is given in Appendix A. This implies that minimizing $\mathcal{L}_\lambda(\phi^\star, \theta)$ is equivalent to minimizing $\mathrm{JS}_\lambda(\rho_{\pi_E}, \rho_{\pi_N}, \rho_{\pi_\theta})$ up to a constant. This relation is similar to the one between binary classification and the JS divergence in GAIL, where a minimax formulation in Eq.(3) is used for occupancy measure matching.

However, solving the minimax problem, $\min_\theta \max_\phi \mathcal{L}_\lambda(\phi, \theta)$, is inappropriate in our method since the JS divergence in Eq.(6) is minimized by $\rho_{\pi_\theta} = (\rho_{\pi_E} + \lambda\rho_{\pi_N})/(1 + \lambda)$ instead by $\rho_{\pi_\theta} = \rho_{\pi_E}$. This minimizer corresponds to a policy mixture, $\pi_\theta(\mathbf{a}|\mathbf{s}) = (\pi_E(\mathbf{a}|\mathbf{s})\rho_{\pi_E}(\mathbf{s}) + \lambda\pi_N(\mathbf{a}|\mathbf{s})\rho_{\pi_N}(\mathbf{s}))/(\rho_{\pi_E}(\mathbf{s}) + \lambda\rho_{\pi_N}(\mathbf{s}))$, which has a positive probability depending on $\lambda$ to select non-expert actions instead of expert actions. Therefore, the above minimax formulation does not lead to occupancy measure matching in our method.

Here, we propose to perform occupancy measure matching by directly minimizing an approximation of $\mathrm{JS}(\rho_{\pi_E}, \rho_{\pi_\theta})$. We can approximate this divergence based on the fact that with $\lambda = 0$, we have $\mathrm{JS}_0(\rho_{\pi_E}, \rho_{\pi_N}, \rho_{\pi_\theta}) = \mathrm{JS}(\rho_{\pi_E}, \rho_{\pi_\theta})$. Since minimizing $\mathcal{L}_\lambda(\phi^\star, \theta)$ is equivalent to minimizing $\mathrm{JS}_\lambda(\rho_{\pi_E}, \rho_{\pi_N}, \rho_{\pi_\theta})$ for any $\lambda$, we can perform occupancy measure matching by solving $\min_\theta \mathcal{L}_0(\phi^\star, \theta)$ with $\lambda = 0$. In practice, we instead minimize an approximation of the divergence based on an intermediate solution $\phi$ since the discriminators do not have infinite capacity and finding a global optimum $\phi^\star$ is not possible in practice. By ignoring constant terms in $\min_\theta \mathcal{L}_0(\phi, \theta)$ and including a causal entropy regularization with $\beta \geqslant 0$, we obtain the following minimization problem for learning $\theta$:

$$\min_\theta \mathbb{E}_{\pi_\theta}\left[\log \frac{\exp(h_\phi(\mathbf{s}, \mathbf{a}))}{\exp(f_\phi(\mathbf{s}, \mathbf{a})) + \exp(h_\phi(\mathbf{s}, \mathbf{a}))}\right] - \beta\mathcal{H}(\pi_\theta). \quad (8)$$

This minimization problem is equivalent to solving a maximum entropy RL problem with reward function $r(\mathbf{s}, \mathbf{a}) = -\log H_\phi(\mathbf{s}, \mathbf{a})$. Eq.(8) also resembles the minimization problem of GAIL in Eq.(3). The main difference between them is that we learn discriminator $H_\phi(\mathbf{s}, \mathbf{a})$ by solving a multiclass classification problem, while GAIL learns discriminator $D_\phi(\mathbf{s}, \mathbf{a})$ by solving a binary classification problem. Due to this similarity, we name our method *multiclass GAIL* (M-GAIL).

The algorithmic summary of our method is given in Appendix C. Our method alternates between updating $\phi$ for maximization using an estimated gradient of $\mathcal{L}_\lambda(\phi, \theta)$ with $\lambda > 0$, and updating $\theta$ for minimization using an estimated gradient of $\mathbb{E}_{\pi_\theta}[\log H_\phi(\mathbf{s}, \mathbf{a})] - \beta\mathcal{H}(\pi_\theta)$. The computation complexity of our method is similar to GAIL, with a small additional cost of computing gradients of $G_\phi(\mathbf{s}, \mathbf{a})$ in the discriminator learning step.

### 4.4 Learning Mixture Policy with Multiclass Classifier

As we have shown, we ensure that M-GAIL learns the expert policy and not a mixture policy by setting $\lambda$ to zero during policy learning. However, learning a mixture policy may be beneficial when selecting non-expert actions with a small probability is not catastrophic. This is because a mixture policy can be learned quickly and accurately thanks to a large number of non-expert demonstrations.

To learn a mixture policy in M-GAIL, we may simply minimize $\mathcal{L}_\lambda(\phi, \theta)$. However, it is unnecessary to use the same $\lambda$ to determine the mixing coefficient of a mixture policy. In particular, we consider *mixture M-GAIL* (MM-GAIL) which learns a multiclass classifier by maximizing $\mathcal{L}_\lambda(\phi, \theta)$ w.r.t. $\phi$ and learns a parameterized policy by solving

$$\min_{\boldsymbol{\theta}} \mathbb{E}_{\pi_{\boldsymbol{\theta}}} \left[ \log \frac{\exp(h_\phi(\mathbf{s}, \mathbf{a}))}{\exp(f_\phi(\mathbf{s}, \mathbf{a})) + \omega \exp(g_\phi(\mathbf{s}, \mathbf{a})) + \exp(h_\phi(\mathbf{s}, \mathbf{a}))} \right] - \beta \mathcal{H}(\pi_{\boldsymbol{\theta}}). \tag{9}$$

Here, the weight $\lambda$ control the influence of non-expert demonstrations during discriminator learning while the weight $0 < \omega < 1$ controls that during policy learning. Using different weights allows us to flexibly control the influence of non-expert demonstrations. For example, by using a large $\lambda$ and a small $\omega$, non-expert demonstrations have high influence during discriminator learning while there is only a small probability of choosing non-expert actions. Note that MM-GAIL does not exactly solve IL problem since it does not learn the expert policy. However, it may have better empirical performance than M-GAIL in tasks where non-expert policy can perform the task moderately well.

## 5 Discussion on Binary Classification with Non-expert Data

Our key idea in this paper is to perform multiclass classification where non-expert demonstrations are regarded as belong to an extra class. A natural question that follows is, can we utilize non-expert demonstrations without considering an extra class? To answer this question, we consider two extensions of GAIL that perform binary classification where non-expert demonstrations are regarded as belong to either expert or agent classes. As shown below, these methods utilize non-expert demonstrations but they do not directly perform IL since they learn a mixture policy.

### 5.1 Non-expert Trajectories as Expert Trajectories

Firstly, we consider an extension of GAIL where we treat non-expert trajectories as expert trajectories and weight their influence by $0 < \lambda < 1$. This can be formalized by an objective function

$$\begin{aligned}
\mathcal{U}_\lambda(\phi, \theta) &= \mathbb{E}_{\pi_{\mathrm{E}}} \left[ \log F_\phi(\mathbf{s}, \mathbf{a}) \right] + \lambda \mathbb{E}_{\pi_{\mathrm{N}}} \left[ \log F_\phi(\mathbf{s}, \mathbf{a}) \right] + \mathbb{E}_{\pi_{\boldsymbol{\theta}}} \left[ \log H_\phi(\mathbf{s}, \mathbf{a}) \right] \\
&= \mathbb{E}_{\pi_{\mathrm{E}} + \lambda \pi_{\mathrm{N}}} \left[ \log F_\phi(\mathbf{s}, \mathbf{a}) \right] + \mathbb{E}_{\pi_{\boldsymbol{\theta}}} \left[ \log H_\phi(\mathbf{s}, \mathbf{a}) \right],
\end{aligned} \tag{10}$$

where $F_\phi(\mathbf{s}, \mathbf{a}) = \exp(f_\phi(\mathbf{s}, \mathbf{a}))/Z_\phi(\mathbf{s}, \mathbf{a})$ and $H_\phi(\mathbf{s}, \mathbf{a}) = \exp(h_\phi(\mathbf{s}, \mathbf{a}))/Z_\phi(\mathbf{s}, \mathbf{a})$ are discriminators with a normalization factor $Z_\phi(\mathbf{s}, \mathbf{a}) = \exp(f_\phi(\mathbf{s}, \mathbf{a})) + \exp(h_\phi(\mathbf{s}, \mathbf{a}))$. Maximizing $\mathcal{U}_\lambda(\phi, \theta)$ w.r.t. $\phi$ is equivalent to learning a binary classifier that classifies between agent's trajectories and a mixture of expert and non-expert trajectories with a mixture coefficient of $1$ and $\lambda$, respectively. We call a method that solves a minimax problem, $\min_{\theta} \max_{\phi} \mathcal{U}_\lambda(\phi, \theta)$, $\mathcal{U}$-GAIL. Notice that, by using the equivalence between binary classification in GAIL and maximum entropy IRL of Fu et al. (2018), learning a binary classification here can be regarded as learning a reward function which is maximized by a mixture of trajectories using maximum entropy IRL.

The auxiliary loss term, $\lambda \mathbb{E}_{\pi_{\mathrm{N}}} \left[ \log F_\phi(\mathbf{s}, \mathbf{a}) \right]$, allows $\mathcal{U}$-GAIL to leverage non-expert demonstrations to improve discriminator learning. However, one important issue of $\mathcal{U}$-GAIL is that it learns a mixture policy instead of the expert policy. To verify this, notice that the optimal discriminator that maximizes $\mathcal{U}_\lambda(\phi, \theta)$ yields a JS divergence: $\mathcal{U}_\lambda(\phi^\star, \theta) = 2\mathrm{JS}(\rho_{\pi_{\mathrm{E}}} + \lambda \rho_{\pi_{\mathrm{N}}}, \rho_{\pi_{\boldsymbol{\theta}}}) - \log 4$, which is minimized by $\rho_{\pi_{\boldsymbol{\theta}}}(\mathbf{s}, \mathbf{a}) = \rho_{\pi_{\mathrm{E}}}(\mathbf{s}, \mathbf{a}) + \lambda \rho_{\pi_{\mathrm{N}}}(\mathbf{s}, \mathbf{a})$. As shown previously, this occupancy measure corresponds to a policy mixture that may select non-expert actions instead of expert actions with a probability monotonically depends on $\lambda$. For this reason, solving $\min_{\theta} \max_{\phi} \mathcal{U}_\lambda(\phi, \theta)$ leads to a mixture policy and does not perform IL.

Recall that minimizing $\mathcal{L}_\lambda(\phi, \theta)$ also leads to the same issue. In M-GAIL, we avoid learning a mixture policy by setting $\lambda$ to zero during policy learning to directly obtain an estimate of $\mathrm{JS}(\rho_{\pi_{\mathrm{E}}}, \rho_{\pi_{\boldsymbol{\theta}}})$ from an estimate of $\mathrm{JS}_\lambda(\rho_{\pi_{\mathrm{E}}}, \rho_{\pi_{\mathrm{N}}}, \rho_{\pi_{\boldsymbol{\theta}}})$. We cannot do this for $\mathcal{U}$-GAIL since the policy learning

objective does not contain $\lambda$ and we cannot change the influence of non-expert demonstrations after the discriminator is learned. Instead, to obtain an estimate of $\mathrm{JS}(\rho_{\pi_E}, \rho_{\pi_\theta})$, it is necessary to find a maximizer of $\mathcal{U}_0(\phi, \theta)$ which implies ignoring non-expert demonstrations during discriminator learning. While we may gradually anneal the value of $\lambda$ to zero to ensure that $\mathcal{U}$-GAIL learns the expert policy in a limit, this may lead to more instability during learning since the optimal discriminator depends on $\lambda$ and $\pi_\theta$ which change during learning.

Another issue of $\mathcal{U}$-GAIL is that, it treats non-expert and expert demonstrations as the same class which implies that the discriminator learns a feature map such that the two datasets are close to each other in the feature space. However, the learned feature map would be less informative when expert and non-expert policies behave too differently and expert and non-expert demonstrations lie far apart in the state-action space.

## 5.2 Non-expert Trajectories as Agent's Trajectories

Alternatively, we may treat non-expert trajectories as trajectories collected by the agent and weight them by $0 < \lambda < 1$. This approach can be formalized via the following objective function:

$$
\begin{aligned}
\mathcal{V}_\lambda(\phi, \theta) &= \mathbb{E}_{\pi_E}\left[\log F_\phi(\mathbf{s}, \mathbf{a})\right] + \lambda\mathbb{E}_{\pi_N}\left[\log H_\phi(\mathbf{s}, \mathbf{a})\right] + \mathbb{E}_{\pi_\theta}\left[\log H_\phi(\mathbf{s}, \mathbf{a})\right] \\
&= \mathbb{E}_{\pi_E}\left[\log F_\phi(\mathbf{s}, \mathbf{a})\right] + \mathbb{E}_{\pi_\theta + \lambda\pi_N}\left[\log H_\phi(\mathbf{s}, \mathbf{a})\right],
\end{aligned}
\tag{11}
$$

where $F_\phi(\mathbf{s}, \mathbf{a})$ and $H_\phi(\mathbf{s}, \mathbf{a})$ are defined identically to those in Eq.(10). Solving a maximization problem, $\max_\phi \mathcal{V}_\lambda(\phi, \theta)$ is equivalent to learning a binary classifier that classifies between expert trajectories and a mixture of agent's and non-expert trajectories with a mixture coefficient of $1$ and $\lambda$, respectively. We call a method that solves the minimax problem, $\min_\theta \max_\phi \mathcal{V}_\lambda(\phi, \theta)$, $\mathcal{V}$-GAIL.

Similarly to $\mathcal{U}$-GAIL, this method learns a mixture policy and does not perform IL. Moreover, it requires an additional constraint to ensure validity of the resulting occupancy measure. To verify these, notice that the optimal discriminator yields $\mathcal{V}_\lambda(\phi^\star, \theta) = 2\mathrm{JS}(\rho_{\pi_E}, \rho_{\pi_\theta} + \lambda\rho_{\pi_N}) - \log 4$, which is minimized by $\rho_{\pi_\theta}(\mathbf{s}, \mathbf{a}) = \rho_{\pi_E}(\mathbf{s}, \mathbf{a}) - \lambda\rho_{\pi_N}(\mathbf{s}, \mathbf{a})$. Since $\lambda > 0$, this optimal occupancy measure may be negative for some $\lambda$ and is not a valid occupancy measure. Thus, $\mathcal{V}$-GAIL requires an additional constraint to ensure non-negativity of the occupancy measure which is not trivial in deep RL. However, in our experiments, $\mathcal{V}$-GAIL can learn good policies without such an explicit constraint. This is likely because non-negativity is implicitly constrained during policy learning by our choice of a Gaussian parameterized policy. However, a relationship between this non-negativity constraint for policy and that for occupancy measure is currently unclear.

## 6 Experiments

We evaluate learning performance of M-GAIL against that of GAIL, $\mathcal{U}$-GAIL and $\mathcal{V}$-GAIL on four continuous control tasks from OpenAI gym (Brockman et al., 2016) with the PyBullet physics simulator (Coumans & Bai, 2018). The discriminator and policy are neural networks with the same architecture used by Ho & Ermon (2016). We use Adam (Kingma & Ba, 2014) with mini-batch size 64 to optimize the discriminator, and we use TRPO (Schulman et al., 2016) to optimize the policy where in each iteration the agent collects $K$ transition samples. For each task, an expert dataset of size $N$ are collected by executing a pre-trained TRPO expert agent. Then, we collect two non-expert datasets by executing two policies which achieve approximately $50\%$ and $70\%$ average returns of the expert agent. The two non-expert datasets consist of $M = 100000$ transition samples and we denote them by $\mathcal{D}_N^{50\%}$ and $\mathcal{D}_N^{70\%}$. We consider $\lambda \in \{0.01, 0.1, 0.5\}$ for M-GAIL, $\mathcal{U}$-GAIL and $\mathcal{V}$-GAIL. More details on hyper-parameter settings and dataset collection are provided in Appendix D.

**Small number of samples scenario:** Firstly, we demonstrate the usefulness of using non-expert demonstrations when the number of expert demonstrations and transition sample collected by the agent are small; $N \in \{100, 300, 500\}$ and $K = 1000$. We also include MM-GAIL which learns a mixture policy in this evaluation. Figure 1 shows the learning curves for $N = 100$ where $\lambda$ is chosen for each method based on the best overall value across tasks[4]. Learning curves for other values of $\lambda$ and $N$ are presented in Appendix E. The result shows that M-GAIL, MM-GAIL and $\mathcal{U}$-GAIL

---

[4]Based on the total return provided in Table 1, we choose $\lambda = 0.01$ for $\mathcal{U}$-GAIL and $\mathcal{V}$-GAIL and $\lambda = 0.5$ for M-GAIL. The values $\lambda = 0.5$ and $\omega = 0.01$ for MM-GAIL are chosen based on M-GAIL and $\mathcal{U}$-GAIL.

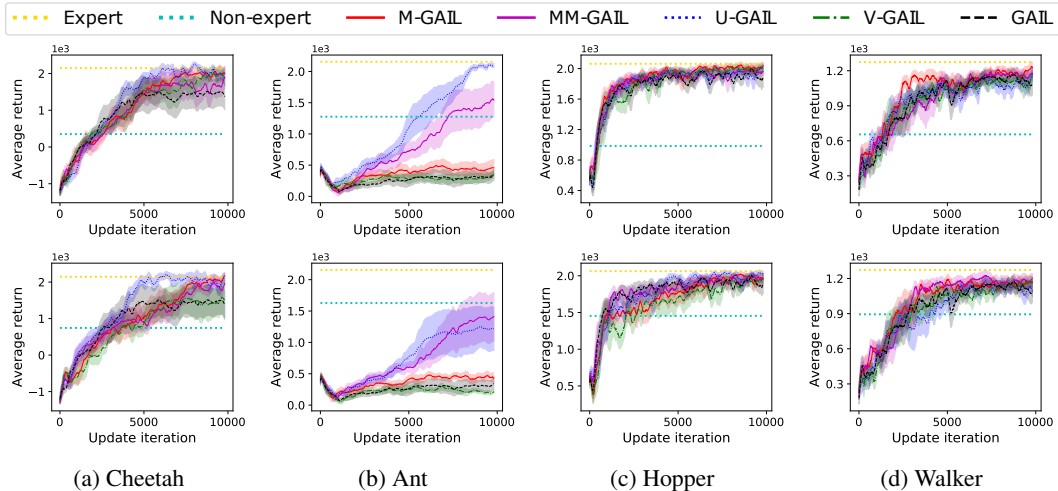

Figure 1: The mean and standard error of average return over 5 trials for $N = 100$ and $K = 1000$ with $\mathcal{D}_{\mathrm{N}}^{50\%}$ (top row) and $\mathcal{D}_{\mathrm{N}}^{70\%}$ (bottom row). The agent collects a total of 10 million samples.

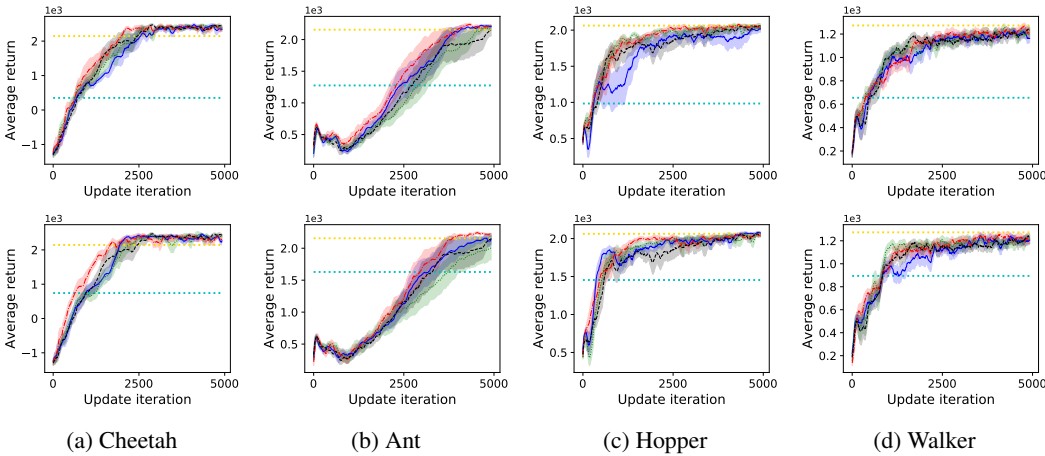

Figure 2: The mean and standard error of average return over 5 trials for $N = 1000$ and $K = 2000$ with $\mathcal{D}_{\mathrm{N}}^{50\%}$ (top row) and $\mathcal{D}_{\mathrm{N}}^{70\%}$ (bottom row). The agent collects a total of 10 million samples.

learn faster and achieve better final returns than that of GAIL, except on Hopper where GAIL also performs quite well. The improvement is significantly large on Cheetah and Ant where GAIL only learns a sub-optimal policy with a relatively large standard error. This instability in GAIL is likely due to learning from a small number of samples.

The result on Ant also shows that $\mathcal{U}$-GAIL tends to outperform M-GAIL. Our conjecture is that non-expert policies in Ant can perform the task moderately well and learning a mixture policy greatly accelerates learning. This is evidenced by good performance of MM-GAIL which also learns a mixture policy. Learning a mixture policy in $\mathcal{U}$-GAIL and MM-GAIL becomes an issue on Walker with $\mathcal{D}_{\mathrm{N}}^{50\%}$. We found that there is a big difference between expert and non-expert actions in this task since the expert agent moves forward while the non-expert agent moves backwards slightly[5]. By learning a mixture of such different policies (moving forward and backward), both $\mathcal{U}$-GAIL and MM-GAIL learn much slower and achieve worse final performance when compared to M-GAIL.

On the other hand, there is no clear difference between $\mathcal{V}$-GAIL and GAIL. This is likely because the expectation for $F_\phi$ in $\mathcal{V}$-GAIL is still approximated by a small number of expert demonstrations

---

[5] The non-expert agent still obtains a moderate average return of 655 since it receives rewards by not falling.

in Eq.(11), and there is also not much benefit in using non-expert demonstrations for $H_\phi$ since the total number of agent trajectories increases as learning progresses.

We also consider the total returns, computed as an averaged area under learning curves, as an evaluation metric for comparing the overall performance given the same number of update iteration. The total return in Table 1 in appendix shows that the overall performance across different values of $\lambda$ of M-GAIL, MM-GAIL and $\mathcal{U}$-GAIL are comparable and they significantly outperform GAIL. The final return in Table 2, computed as average returns over the last 500 iterations, further confirms that using non-expert demonstrations leads to a better final policy when compared to that of GAIL.

**Large number of samples scenario:** Next, we consider a scenario with relatively larger numbers of expert demonstrations and collected transition samples; $N \in \{1000, 10000\}$ and $K = 2000$. Figure 2 shows the learning curves for $N = 1000$ for the same $\lambda$ as the previous scenario. The result shows that using non-expert demonstrations improve the performance in the early stage of learning when the agent have not yet collected many samples. However, the improvement is smaller when compared to the previous scenario as expected since a large number of expert demonstrations already provide sufficient information to learn an accurate binary classifier in GAIL.

We can also see that $\mathcal{U}$-GAIL performs rather poorly when compared to M-GAIL, especially on Hopper with $\mathcal{D}_N^{50\%}$. Our conjecture is that $\mathcal{U}$-GAIL improves learning by biasing a binary classifier using the non-expert auxiliary loss term. However, when the binary classifier can be learned sufficiently accurately by expert demonstrations alone, the bias introduced would instead degrade the classifier and not improve it. In contrast, M-GAIL always classifies between the three classes and does not use bias to improve learning. This result further shows that our multiclass classification approach is more robust to the choice of non-expert policy when compared to approaches based on binary classification.

The total return in Table 3 shows that M-GAIL overall performs better across different values of $\lambda$ and tasks when compared to other methods, except on Ant where $\mathcal{U}$-GAIL performs the best overall. However, we can also see in Table 4 that differences in the final returns are quite small and all methods achieve similar final performance. Nonetheless, performance improvement in the early stage of learning still confirms that non-expert demonstrations improves discriminator learning in GAIL.

## 7 CONCLUSION

We presented M-GAIL, a method that improves GAIL by using non-expert demonstration as an extra class in discriminator learning. Compared to related methods that use an additional dataset for IL, M-GAIL relies on a less restrictive assumption on the dataset and can efficiently train deep neural networks. Using an extra class also avoids learning a mixture policy which is an issue of GAIL extensions that directly include non-expert demonstrations into existing datasets. On the other hand, when learning a mixture policy is beneficial, our mixture M-GAIL can also learn a mixture policy with a benefit of more flexibility in choosing a mixture coefficient.

The recent result of Fu et al. (2018) shows that binary classification in GAIL and reward learning in maximum entropy IRL are equivalent. This suggests that there is such equivalence for multiclass classification. However, as we show in Appendix B, this is not the case and our method does not learn a reward function. Since IRL is also a promising approach to learn an expert policy, developing a deep IRL method that utilizes non-expert demonstrations is a promising research direction.

Our method can also be extended to use trajectories collected by the agent's previous policies as an extra class. This approach allows us to reuse trajectories collected in the previous iterations without relying on importance weight which has high variance. However, having two changing classes may greatly destabilize learning and a stabilization method would be required.

We developed M-GAIL based on JS-divergence. However, recent research suggests that other distances such as Wasserstein distance are more suitable when learning from image data (Arjovsky et al., 2017). Extending our method to use such distances is an important future work for applying our idea to solve image-based tasks such as Atari games (Mnih et al., 2015) and autonomous driving (Li et al., 2017b).

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

## A  MAXIMUM OF WEIGHT LOG-LIKELIHOOD OBJECTIVE APPROXIMATES JENSEN-SHANNON DIVERGENCE

If we assume that the functions $F_\phi$, $G_\phi$, and $H_\phi$ are non-parametric and have infinite capacity, we can rewrite the multiclass classification objective as

$$\mathcal{L}_\lambda(F, G, H) = \mathbb{E}_{\pi_\mathrm{E}}\left[\log F(\mathbf{s}, \mathbf{a})\right] + \lambda\mathbb{E}_{\pi_\mathrm{N}}\left[\log G(\mathbf{s}, \mathbf{a})\right] + \mathbb{E}_{\pi_\theta}\left[\log H(\mathbf{s}, \mathbf{a})\right]. \tag{12}$$

Recall that $\mathbb{E}_\pi\left[r(\mathbf{s}, \mathbf{a})\right] = \iint \rho_\pi(\mathbf{s}, \mathbf{a})r(\mathbf{s}, \mathbf{a})\mathrm{d}\mathbf{s}\mathrm{d}\mathbf{a}$ where $\rho_\pi$ is the occupancy measure of $\pi$. This objective is a concave function and we can find its global maximizer under constraints $F(\mathbf{s}, \mathbf{a}) \geqslant 0$, $G(\mathbf{s}, \mathbf{a}) \geqslant 0$, $H(\mathbf{s}, \mathbf{a}) \geqslant 0$ and $F(\mathbf{s}, \mathbf{a}) + G(\mathbf{s}, \mathbf{a}) + H(\mathbf{s}, \mathbf{a}) = 1, \forall\mathbf{s}\forall\mathbf{a}$, by using the method of Lagrange multipliers. The global maximizer corresponds to

$$F^\star(\mathbf{s}, \mathbf{a}) = \frac{\rho_{\pi_\mathrm{E}}(\mathbf{s}, \mathbf{a})}{q(\mathbf{s}, \mathbf{a})}, \quad G^\star(\mathbf{s}, \mathbf{a}) = \frac{\lambda\rho_{\pi_\mathrm{N}}(\mathbf{s}, \mathbf{a})}{q(\mathbf{s}, \mathbf{a})}, \quad H^\star(\mathbf{s}, \mathbf{a}) = \frac{\rho_{\pi_\theta}(\mathbf{s}, \mathbf{a})}{q(\mathbf{s}, \mathbf{a})}, \tag{13}$$

where $q(\mathbf{s}, \mathbf{a}) = \rho_{\pi_\mathrm{E}}(\mathbf{s}, \mathbf{a}) + \lambda\rho_{\pi_\mathrm{N}}(\mathbf{s}, \mathbf{a}) + \rho_{\pi_\theta}(\mathbf{s}, \mathbf{a})$. Let $\bar{q}(\mathbf{s}, \mathbf{a}) = \frac{q(\mathbf{s},\mathbf{a})}{2+\lambda}$, by substituting these quantities back to $\mathcal{L}_\lambda$, we obtain the maximum objective value as

$$\begin{aligned}
\mathcal{L}_\lambda(F^\star, G^\star, H^\star) &= \mathbb{E}_{\pi_\mathrm{E}}\left[\log\frac{\rho_{\pi_\mathrm{E}}(\mathbf{s}, \mathbf{a})}{q(\mathbf{s}, \mathbf{a})}\right] + \lambda\mathbb{E}_{\pi_\mathrm{N}}\left[\log\frac{\lambda\rho_{\pi_\mathrm{N}}(\mathbf{s}, \mathbf{a})}{q(\mathbf{s}, \mathbf{a})}\right] + \mathbb{E}_{\pi_\theta}\left[\log\frac{\rho_{\pi_\theta}(\mathbf{s}, \mathbf{a})}{q(\mathbf{s}, \mathbf{a})}\right] \\
&= \mathbb{E}_{\pi_\mathrm{E}}\left[\log\frac{\rho_{\pi_\mathrm{E}}(\mathbf{s}, \mathbf{a})}{\bar{q}(\mathbf{s}, \mathbf{a})}\right] + \lambda\mathbb{E}_{\pi_\mathrm{N}}\left[\log\frac{\rho_{\pi_\mathrm{N}}(\mathbf{s}, \mathbf{a})}{\bar{q}(\mathbf{s}, \mathbf{a})}\right] + \mathbb{E}_{\pi_\theta}\left[\log\frac{\rho_{\pi_\theta}(\mathbf{s}, \mathbf{a})}{\bar{q}(\mathbf{s}, \mathbf{a})}\right] \\
&\quad + \lambda\log\lambda - (2 + \lambda)\log(2 + \lambda). \tag{14}
\end{aligned}$$

Notice that the first, second, and third terms look similar to Kullback-Leibler (KL) divergences which are defined as $\mathrm{KL}(p_1||p_2) = \iint p_1(\mathbf{s}, \mathbf{a})\log\frac{p_1(\mathbf{s},\mathbf{a})}{p_1(\mathbf{s},\mathbf{a})}\mathrm{d}\mathbf{s}\mathrm{d}\mathbf{a}$, for probability densities $p_1$ and $p_2$. However, the occupancy measures are unnormalized densities and theoretically we cannot use KL divergences to rewrite Eq.(14). For unnormalized densities $\rho(\mathbf{s}, \mathbf{a})$ and $\bar{q}(\mathbf{s}, \mathbf{a})$, we consider a generalized KL divergence defined as

$$\begin{aligned}
\mathrm{gKL}(\rho||\bar{q}) &= \iint \rho(\mathbf{s}, \mathbf{a})\log\frac{\rho(\mathbf{s}, \mathbf{a})}{\bar{q}(\mathbf{s}, \mathbf{a})}\mathrm{d}\mathbf{s}\mathrm{d}\mathbf{a} - \iint \rho(\mathbf{s}, \mathbf{a})\mathrm{d}\mathbf{s}\mathrm{d}\mathbf{a} + \iint \bar{q}(\mathbf{s}, \mathbf{a})\mathrm{d}\mathbf{s}\mathrm{d}\mathbf{a} \\
&= \mathrm{KL}(\rho||\bar{q}) - \iint \rho(\mathbf{s}, \mathbf{a})\mathrm{d}\mathbf{s}\mathrm{d}\mathbf{a} + \iint \bar{q}(\mathbf{s}, \mathbf{a})\mathrm{d}\mathbf{s}\mathrm{d}\mathbf{a}. \tag{15}
\end{aligned}$$

We can show that for Eq.(14), gKL and KL are equivalent since the extra terms cancel out:

$$\begin{aligned}
\mathrm{gKL}(\rho_{\pi_\mathrm{E}}||\bar{q}) &+ \lambda\mathrm{gKL}(\rho_{\pi_\mathrm{N}}||\bar{q}) + \mathrm{gKL}(\rho_{\pi_\theta}||\bar{q}) \\
&= \mathrm{KL}(\rho_{\pi_\mathrm{E}}||\bar{q}) + \lambda\mathrm{KL}(\rho_{\pi_\mathrm{N}}||\bar{q}) + \mathrm{KL}(\rho_{\pi_\theta}||\bar{q}) \\
&\quad - \iint\left(\rho_{\pi_\mathrm{E}}(\mathbf{s}, \mathbf{a}) + \lambda\rho_{\pi_\mathrm{N}}(\mathbf{s}, \mathbf{a}) + \rho_{\pi_\theta}(\mathbf{s}, \mathbf{a})\right)\mathrm{d}\mathbf{s}\mathrm{d}\mathbf{a} + (2 + \lambda)\iint \bar{q}(\mathbf{s}, \mathbf{a})\mathrm{d}\mathbf{s}\mathrm{d}\mathbf{a} \\
&= \mathrm{KL}(\rho_{\pi_\mathrm{E}}||\bar{q}) + \lambda\mathrm{KL}(\rho_{\pi_\mathrm{N}}||\bar{q}) + \mathrm{KL}(\rho_{\pi_\theta}||\bar{q}) \\
&\quad - \iint\left(\rho_{\pi_\mathrm{E}}(\mathbf{s}, \mathbf{a}) + \lambda\rho_{\pi_\mathrm{N}}(\mathbf{s}, \mathbf{a}) + \rho_{\pi_\theta}(\mathbf{s}, \mathbf{a})\right)\mathrm{d}\mathbf{s}\mathrm{d}\mathbf{a} + \iint q(\mathbf{s}, \mathbf{a})\mathrm{d}\mathbf{s}\mathrm{d}\mathbf{a} \\
&= \mathrm{KL}(\rho_{\pi_\mathrm{E}}||\bar{q}) + \lambda\mathrm{KL}(\rho_{\pi_\mathrm{N}}||\bar{q}) + \mathrm{KL}(\rho_{\pi_\theta}||\bar{q}). \tag{16}
\end{aligned}$$

Then, the optimal value in Eq.(14) can be rewritten in terms of gKL divergences as

$$\begin{aligned}
\mathcal{L}_\lambda(F^\star, G^\star, H^\star) &= \mathrm{gKL}(\rho_{\pi_\mathrm{E}}||\bar{q}) + \lambda\mathrm{gKL}(\rho_{\pi_\mathrm{N}}||\bar{q}) + \mathrm{gKL}(\rho_{\pi_\theta}||\bar{q}) \\
&\quad + \lambda\log\lambda - (2 + \lambda)\log(2 + \lambda) \tag{17}
\end{aligned}$$

Next, we consider a JS divergence among $\rho_{\pi_\mathrm{E}}$, $\rho_{\pi_\mathrm{N}}$, and $\rho_{\pi_\theta}$ with corresponding weights $1/(2 + \lambda)$, $\lambda/(2 + \lambda)$, and $1/(2 + \lambda)$, which can be defined based on (generalized) KL divergence as

$$\mathrm{JS}_\lambda(\rho_{\pi_\mathrm{E}}, \rho_{\pi_\mathrm{N}}, \rho_{\pi_\theta}) = \frac{1}{2 + \lambda}\left(\mathrm{gKL}(\rho_{\pi_\mathrm{E}}||\bar{q}) + \lambda\mathrm{gKL}(\rho_{\pi_\mathrm{N}}||\bar{q}) + \mathrm{gKL}(\rho_{\pi_\theta}||\bar{q})\right). \tag{18}$$

Note that we use the fact that the sum of gKL and the sum of KL are equivalent to define the JS divergence. Alternatively, we may use a definition of JS in terms of a Shannon entropy,

$JS_\lambda(\rho_{\pi_E}, \rho_{\pi_N}, \rho_{\pi_\theta}) = -\frac{1}{2+\lambda}\left(H(\rho_{\pi_E}) + \lambda H(\rho_{\pi_N}) + H(\rho_{\pi_\theta})\right) + H\left(\frac{1}{2+\lambda}\left(\rho_{\pi_E} + \lambda\rho_{\pi_N} + \rho_{\pi_\theta}\right)\right),$
where $H(\rho) = -\iint \rho_\pi(\mathbf{s}, \mathbf{a})\log\rho_\pi(\mathbf{s}, \mathbf{a})\mathrm{d}\mathbf{s}\mathrm{d}\mathbf{a}$, to obtain the same definition of JS divergence.

By comparing Eq.(17) and Eq.(18), we have that the maximum value of $\mathcal{L}_\lambda$ corresponds to $JS_\lambda$ with a positive scaling and a constant shift:

$$\mathcal{L}_\lambda(F^\star, G^\star, H^\star) = (2 + \lambda)JS_\lambda(\rho_{\pi_E}, \rho_{\pi_N}, \rho_{\pi_\theta}) + \log\left(\lambda^\lambda(2 + \lambda)^{-(2+\lambda)}\right). \tag{19}$$

Notice that $JS_\lambda(\rho_{\pi_E}, \rho_{\pi_N}, \rho_{\pi_\theta})$ is a convex function of $\rho_{\pi_\theta}$. By setting its derivative to zero, we have that the minimum of $JS_\lambda(\rho_{\pi_E}, \rho_{\pi_N}, \rho_{\pi_\theta})$ is obtained when $\rho_{\pi_\theta} = \frac{\rho_{\pi_E} + \lambda\rho_{\pi_N}}{1+\lambda}$. Then, the corresponding mixture policy can be obtained by using the one-to-one correspondence between occupancy measure and policy given as $\pi(\mathbf{a}|\mathbf{s}) = \frac{\rho_\pi(\mathbf{s},\mathbf{a})}{\rho_\pi(\mathbf{s})}$.

# B  RELATION TO MAXIMUM ENTROPY INVERSE REINFORCEMENT LEARNING

Recently, Finn et al. (2016a) and Fu et al. (2018) showed an interesting result that an objective for solving a binary classification problem in generative adversarial training is related to a maximum entropy IRL objective (Ziebart et al., 2008; 2010). Here, we show that multiclass classification objective only approximates a maximum entropy IRL objective with two reward functions.

Let us consider the goal of learning the expert reward function $r_E(\mathbf{s}, \mathbf{a})$ and the non-expert reward function $r_N(\mathbf{s}, \mathbf{a})$ by parameterized functions $f_\phi(\mathbf{s}, \mathbf{a})$, and $g_\phi(\mathbf{s}, \mathbf{a})$, respectively. $r_N(\mathbf{s}, \mathbf{a})$ is a reward function whose $\pi_N$ is optimal[6]. The parameter $\phi$ of the two reward functions may be learned independently using maximum entropy IRL. However, we are interested in jointly learn both reward functions by maximizing the following objective:

$$\mathcal{M}_\lambda(\phi) = \mathbb{E}_{p_E(\tau)}\left[\log p_f(\tau)\right] + \lambda\mathbb{E}_{p_N(\tau)}\left[\log p_g(\tau)\right]. \tag{20}$$

The first and second terms are the maximum entropy IRL objectives for learning the expert reward and the non-expert reward, respectively. The weight parameter $\lambda > 0$ controls the influence of the second objective. The trajectory density $p_f(\tau)$ is defined as $p_f(\tau) \propto p_0(\mathbf{s}_0)\prod_{t=0}^{T} p(\mathbf{s}_{t+1}|\mathbf{s}_t, \mathbf{a}_t)\exp(f_\phi(\mathbf{s}_t, \mathbf{a}_t))$ and $p_g(\tau)$ is also defined similarly. The following proposition shows that $\mathcal{L}_\lambda(\phi, \theta)$ is a surrogate of $\mathcal{M}_\lambda(\phi)$.

**Proposition 1.** *Under the assumption that $h_\phi(\mathbf{s}, \mathbf{a}) = \pi_\theta(\mathbf{a}|\mathbf{s})$ for fixed $\theta$, the gradient of $\mathcal{L}_\lambda(\phi, \theta)$ is a biased gradient of $\mathcal{M}_\lambda(\phi)$. The bias vanishes when $\lambda = 0$ and $\pi_\theta$ is an optimal maximum entropy policy for reward function $f_\phi(\mathbf{s}, \mathbf{a})$.*

This proposition indicates that local maxima of $\mathcal{M}_\lambda(\phi)$ are approximated by local maxima of $\mathcal{L}_\lambda(\phi, \theta)$, and that approximation errors decrease as $\lambda$ decreases to zero. Due to approximation errors, our method does not perform IRL and does not learn the true reward functions. However, it still suggests that we may still use $f_\phi(\mathbf{s}, \mathbf{a})$ and $g_\phi(\mathbf{s}, \mathbf{a})$ as approximations of the reward functions which could be useful in the IRLF framework of Shiarlis et al. (2016) when the non-expert demonstrations are clearly failure demonstrations.

The proof of Proposition 1 is an extension of the proof by Fu et al. (2018) for their adversarial IRL (AIRL) method, but with additional terms from non-expert demonstrations in our case. That is, we show that the gradient of a maximum entropy IRL objective proposed by Finn et al. (2016b) with specific importance weights only differs from the gradient of the discriminator learning objective in the importance weights. Similar to the case of AIRL, we only consider the case of non-discounted,

---

[6]The optimal reward function always exists since for any trajectory there is at least one reward function that makes the trajectory optimal (Ng & Russell, 2000).

finite horizon MDP. The gradient of this objective w.r.t. $\boldsymbol{\phi}$ is

$$
\nabla_{\boldsymbol{\phi}} \mathcal{M}_\lambda(\boldsymbol{\phi}) = \mathbb{E}_{p_{\mathrm{E}}(\boldsymbol{\tau})} \left[ \sum_{t=0}^{T} \nabla_{\boldsymbol{\phi}} f_{\boldsymbol{\phi}}(\mathbf{s}_t, \mathbf{a}_t) \right] - \nabla_{\boldsymbol{\phi}} \log \int p_0(\mathbf{s}_0) \prod_{t=0}^{T} p(\mathbf{s}_{t+1}|\mathbf{s}_t, \mathbf{a}_t) \exp(f_{\boldsymbol{\phi}}(\mathbf{s}_t, \mathbf{a}_t)) \mathrm{d}\boldsymbol{\tau}
$$

$$
+ \lambda \mathbb{E}_{p_{\mathrm{N}}(\boldsymbol{\tau})} \left[ \sum_{t=0}^{T} \nabla_{\boldsymbol{\phi}} g_{\boldsymbol{\phi}}(\mathbf{s}_t, \mathbf{a}_t) \right] - \lambda \nabla_{\boldsymbol{\phi}} \log \int p_0(\mathbf{s}_0) \prod_{t=0}^{T} p(\mathbf{s}_{t+1}|\mathbf{s}_t, \mathbf{a}_t) \exp(g_{\boldsymbol{\phi}}(\mathbf{s}_t, \mathbf{a}_t)) \mathrm{d}\boldsymbol{\tau}
$$

$$
= \mathbb{E}_{p_{\mathrm{E}}(\boldsymbol{\tau})} \left[ \sum_{t=0}^{T} \nabla_{\boldsymbol{\phi}} f_{\boldsymbol{\phi}}(\mathbf{s}_t, \mathbf{a}_t) \right] - \mathbb{E}_{p_f(\boldsymbol{\tau})} \left[ \sum_{t=0}^{T} \nabla_{\boldsymbol{\phi}} f_{\boldsymbol{\phi}}(\mathbf{s}_t, \mathbf{a}_t) \right]
$$

$$
+ \lambda \mathbb{E}_{p_{\mathrm{N}}(\boldsymbol{\tau})} \left[ \sum_{t=0}^{T} \nabla_{\boldsymbol{\phi}} g_{\boldsymbol{\phi}}(\mathbf{s}_t, \mathbf{a}_t) \right] - \lambda \mathbb{E}_{p_g(\boldsymbol{\tau})} \left[ \sum_{t=0}^{T} \nabla_{\boldsymbol{\phi}} g_{\boldsymbol{\phi}}(\mathbf{s}_t, \mathbf{a}_t) \right]. \tag{21}
$$

Let $p_{\mathrm{E}}^t(\mathbf{s}, \mathbf{a}) = \int p_{\mathrm{E}}(\boldsymbol{\tau}) \mathrm{d}\boldsymbol{\tau}_{t' \neq t}$ be a state-action marginal density at time $t$ obtained by marginalizing $p_{\mathrm{E}}(\boldsymbol{\tau})$ over states and actions at all time-step except at $t$. The state-action marginal densities $p_{\mathrm{N}}^t(\mathbf{s}, \mathbf{a})$, $p_f^t(\mathbf{s}, \mathbf{a})$, and $p_g^t(\mathbf{s}, \mathbf{a})$ are defined similarly. We can rewrite the expectations over trajectories as a sum of expectations over state-action marginals as follows:

$$
\nabla_{\boldsymbol{\phi}} \mathcal{M}_\lambda(\boldsymbol{\phi}) = \sum_{t=0}^{T} \left\{ \mathbb{E}_{p_{\mathrm{E}}^t(\mathbf{s}, \mathbf{a})} \left[ \nabla_{\boldsymbol{\phi}} f_{\boldsymbol{\phi}}(\mathbf{s}, \mathbf{a}) \right] - \mathbb{E}_{p_f^t(\mathbf{s}, \mathbf{a})} \left[ \nabla_{\boldsymbol{\phi}} f_{\boldsymbol{\phi}}(\mathbf{s}, \mathbf{a}) \right] \right.
$$

$$
\left. + \lambda \mathbb{E}_{p_{\mathrm{N}}^t(\mathbf{s}, \mathbf{a})} \left[ \nabla_{\boldsymbol{\phi}} g_{\boldsymbol{\phi}}(\mathbf{s}, \mathbf{a}) \right] - \lambda \mathbb{E}_{p_g^t(\mathbf{s}, \mathbf{a})} \left[ \nabla_{\boldsymbol{\phi}} g_{\boldsymbol{\phi}}(\mathbf{s}, \mathbf{a}) \right] \right\}. \tag{22}
$$

Let $\mu_{\boldsymbol{\theta}}^t(\mathbf{s}, \mathbf{a}) = p_{\mathrm{E}}^t(\mathbf{s}, \mathbf{a}) + \lambda p_{\mathrm{N}}^t(\mathbf{s}, \mathbf{a}) + p_{\boldsymbol{\theta}}^t(\mathbf{s}, \mathbf{a})$ be a (unnormalized) sampling distribution where $p_{\boldsymbol{\theta}}^t(\mathbf{s}, \mathbf{a})$ is a state-action marginal of a trajectory density $p_{\boldsymbol{\theta}}(\boldsymbol{\tau}) = p_0(\mathbf{s}_0) \prod_{t=0}^{T} p(\mathbf{s}_{t+1}|\mathbf{s}_t, \mathbf{a}_t) \pi_{\boldsymbol{\theta}}(\mathbf{a}_t|\mathbf{s}_t)$. By rearranging terms and using importance weights for the expectations over $p_f^t(\mathbf{s}, \mathbf{a})$ and $p_g^t(\mathbf{s}, \mathbf{a})$, we can rewrite the gradient as

$$
\nabla_{\boldsymbol{\phi}} \mathcal{M}_\lambda(\boldsymbol{\phi}) = \sum_{t=0}^{T} \left\{ \mathbb{E}_{p_{\mathrm{E}}^t(\mathbf{s}, \mathbf{a})} \left[ \nabla_{\boldsymbol{\phi}} f_{\boldsymbol{\phi}}(\mathbf{s}, \mathbf{a}) \right] + \lambda \mathbb{E}_{p_{\mathrm{N}}^t(\mathbf{s}, \mathbf{a})} \left[ \nabla_{\boldsymbol{\phi}} g_{\boldsymbol{\phi}}(\mathbf{s}, \mathbf{a}) \right] \right.
$$

$$
\left. - \mathbb{E}_{\mu_{\boldsymbol{\theta}}^t(\mathbf{s}, \mathbf{a})} \left[ \frac{p_f^t(\mathbf{s}, \mathbf{a})}{\mu_{\boldsymbol{\theta}}^t(\mathbf{s}, \mathbf{a})} \nabla_{\boldsymbol{\phi}} f_{\boldsymbol{\phi}}(\mathbf{s}, \mathbf{a}) + \lambda \frac{p_g^t(\mathbf{s}, \mathbf{a})}{\mu_{\boldsymbol{\theta}}^t(\mathbf{s}, \mathbf{a})} \nabla_{\boldsymbol{\phi}} g_{\boldsymbol{\phi}}(\mathbf{s}, \mathbf{a}) \right] \right\}. \tag{23}
$$

Next, we show that $\nabla_{\boldsymbol{\phi}} \mathcal{L}_\lambda(\boldsymbol{\phi}, \boldsymbol{\theta})$ has the same form as $\nabla_{\boldsymbol{\phi}} \mathcal{M}_\lambda(\boldsymbol{\phi})$ but with incorrect importance weights. First, we rewrite $\mathcal{L}_\lambda(\boldsymbol{\phi}, \boldsymbol{\theta})$ using the state-action marginal densities as

$$
\mathcal{L}_\lambda(\boldsymbol{\phi}, \boldsymbol{\theta}) = \mathbb{E}_{\pi_{\mathrm{E}}} \left[ \log F_{\boldsymbol{\phi}}(\mathbf{s}, \mathbf{a}) \right] + \lambda \mathbb{E}_{\pi_{\mathrm{N}}} \left[ \log G_{\boldsymbol{\phi}}(\mathbf{s}, \mathbf{a}) \right] + \mathbb{E}_{\pi_{\boldsymbol{\theta}}} \left[ \log H_{\boldsymbol{\phi}}(\mathbf{s}, \mathbf{a}) \right]
$$

$$
= \mathbb{E}_{p_{\mathrm{E}}(\boldsymbol{\tau})} \left[ \sum_{t=0}^{T} \log F_{\boldsymbol{\phi}}(\mathbf{s}_t, \mathbf{a}_t) \right] + \lambda \mathbb{E}_{p_{\mathrm{N}}(\boldsymbol{\tau})} \left[ \sum_{t=0}^{T} \log G_{\boldsymbol{\phi}}(\mathbf{s}_t, \mathbf{a}_t) \right] + \mathbb{E}_{p_{\boldsymbol{\theta}}(\boldsymbol{\tau})} \left[ \sum_{t=0}^{T} \log H_{\boldsymbol{\phi}}(\mathbf{s}_t, \mathbf{a}_t) \right]
$$

$$
= \sum_{t=0}^{T} \left\{ \mathbb{E}_{p_{\mathrm{E}}^t(\mathbf{s}, \mathbf{a})} \left[ \log F_{\boldsymbol{\phi}}(\mathbf{s}, \mathbf{a}) \right] + \lambda \mathbb{E}_{p_{\mathrm{N}}^t(\mathbf{s}, \mathbf{a})} \left[ \log G_{\boldsymbol{\phi}}(\mathbf{s}, \mathbf{a}) \right] + \mathbb{E}_{p_{\boldsymbol{\theta}}^t(\mathbf{s}, \mathbf{a})} \left[ \log H_{\boldsymbol{\phi}}(\mathbf{s}, \mathbf{a}) \right] \right\}, \tag{24}
$$

where in the second line we use the definition of the expected return for non-discounted finite-horizon case. Replacing $F_{\boldsymbol{\phi}}$, $G_{\boldsymbol{\phi}}$, and $H_{\boldsymbol{\phi}}$ by their parameterization in Eq.(4) with $h(\mathbf{s}, \mathbf{a}) = \log \pi_{\boldsymbol{\theta}}(\mathbf{a}|\mathbf{s})$ gives

$$
\mathcal{L}_\lambda(\boldsymbol{\phi}, \boldsymbol{\theta}) = \sum_{t=0}^{T} \left\{ \mathbb{E}_{p_{\mathrm{E}}^t(\mathbf{s}, \mathbf{a})} \left[ f_{\boldsymbol{\phi}}(\mathbf{s}, \mathbf{a}) \right] + \lambda \mathbb{E}_{p_{\mathrm{N}}^t(\mathbf{s}, \mathbf{a})} \left[ g_{\boldsymbol{\phi}}(\mathbf{s}, \mathbf{a}) + \log \lambda \right] + \mathbb{E}_{p_{\boldsymbol{\theta}}^t(\mathbf{s}, \mathbf{a})} \left[ \log \pi_{\boldsymbol{\theta}}(\mathbf{a}|\mathbf{s}) \right] \right.
$$

$$
\left. - \mathbb{E}_{p_{\mathrm{E}}^t(\mathbf{s}, \mathbf{a}) + \lambda p_{\mathrm{N}}^t(\mathbf{s}, \mathbf{a}) + p_{\boldsymbol{\theta}}^t(\mathbf{s}, \mathbf{a})} \left[ \log \left( \exp(f_{\boldsymbol{\phi}}(\mathbf{s}, \mathbf{a})) + \lambda \exp(g_{\boldsymbol{\phi}}(\mathbf{s}, \mathbf{a})) + \pi_{\boldsymbol{\theta}}(\mathbf{a}|\mathbf{s}) \right) \right] \right\}
$$

$$
= \sum_{t=0}^{T} \left\{ \mathbb{E}_{p_{\mathrm{E}}^t(\mathbf{s}, \mathbf{a})} \left[ f_{\boldsymbol{\phi}}(\mathbf{s}, \mathbf{a}) \right] + \lambda \mathbb{E}_{p_{\mathrm{N}}^t(\mathbf{s}, \mathbf{a})} \left[ g_{\boldsymbol{\phi}}(\mathbf{s}, \mathbf{a}) + \log \lambda \right] + \mathbb{E}_{p_{\boldsymbol{\theta}}^t(\mathbf{s}, \mathbf{a})} \left[ \log \pi_{\boldsymbol{\theta}}(\mathbf{a}|\mathbf{s}) \right] \right.
$$

$$
\left. - \mathbb{E}_{\mu_{\boldsymbol{\theta}}^t(\mathbf{s}, \mathbf{a})} \left[ \log \left( \exp(f_{\boldsymbol{\phi}}(\mathbf{s}, \mathbf{a})) + \lambda \exp(g_{\boldsymbol{\phi}}(\mathbf{s}, \mathbf{a})) + \pi_{\boldsymbol{\theta}}(\mathbf{a}|\mathbf{s}) \right) \right] \right\}, \tag{25}
$$

where we use $\log(\lambda \exp(g_\phi(\mathbf{s}, \mathbf{a}))) = g_\phi(\mathbf{s}, \mathbf{a}) + \log \lambda$ in the first equality and use $\mu_\theta^t(\mathbf{s}, \mathbf{a})$ defined above in the second equality. Then, the gradient of $\mathcal{L}_\lambda(\phi, \theta)$ w.r.t. $\phi$ is given by

$$
\begin{aligned}
\nabla_\phi \mathcal{L}_\lambda(\phi, \theta) = \sum_{t=0}^T &\Big\{ \mathbb{E}_{p_{\mathrm{E}}^t(\mathbf{s}, \mathbf{a})}\left[\nabla_\phi f_\phi(\mathbf{s}, \mathbf{a})\right] + \lambda \mathbb{E}_{p_{\mathrm{N}}^t(\mathbf{s}, \mathbf{a})}\left[\nabla_\phi g_\phi(\mathbf{s}, \mathbf{a})\right] \\
&- \mathbb{E}_{\mu_\theta^t(\mathbf{s}, \mathbf{a})}\left[\nabla_\phi \log\left(\exp(f_\phi(\mathbf{s}, \mathbf{a})) + \lambda \exp(g_\phi(\mathbf{s}, \mathbf{a})) + \pi_\theta(\mathbf{a}|\mathbf{s})\right)\right] \Big\} \\
= \sum_{t=0}^T &\Big\{ \mathbb{E}_{p_{\mathrm{E}}^t(\mathbf{s}, \mathbf{a})}\left[\nabla_\phi f_\phi(\mathbf{s}, \mathbf{a})\right] + \lambda \mathbb{E}_{p_{\mathrm{N}}^t(\mathbf{s}, \mathbf{a})}\left[\nabla_\phi g_\phi(\mathbf{s}, \mathbf{a})\right] \\
&- \mathbb{E}_{\mu_\theta^t(\mathbf{s}, \mathbf{a})}\left[\frac{\exp(f_\phi(\mathbf{s}, \mathbf{a}))}{Z_\phi(\mathbf{s}, \mathbf{a})}\nabla_\phi f_\phi(\mathbf{s}, \mathbf{a}) + \lambda \frac{\exp(g_\phi(\mathbf{s}, \mathbf{a}))}{Z_\phi(\mathbf{s}, \mathbf{a})}\nabla_\phi g_\phi(\mathbf{s}, \mathbf{a})\right] \Big\},
\end{aligned}
\tag{26}
$$

where $Z_\phi(\mathbf{s}, \mathbf{a}) = \exp(f_\phi(\mathbf{s}, \mathbf{a})) + \lambda \exp(g_\phi(\mathbf{s}, \mathbf{a})) + \pi_\theta(\mathbf{a}|\mathbf{s})$. Let $p_\theta^t(\mathbf{s}) = \int p_\theta^t(\mathbf{s}, \mathbf{a})\mathrm{d}\mathbf{a}$ be a state marginal of the policy trajectory density. By multiplying both the nominators and denominators of the third and fourth terms by $p_\theta^t(\mathbf{s})$, we obtain

$$
\begin{aligned}
\nabla_\phi \mathcal{L}_\lambda(\phi, \theta) = \sum_{t=0}^T &\Big\{ \mathbb{E}_{p_{\mathrm{E}}^t(\mathbf{s}, \mathbf{a})}\left[\nabla_\phi f_\phi(\mathbf{s}, \mathbf{a})\right] + \lambda \mathbb{E}_{p_{\mathrm{N}}^t(\mathbf{s}, \mathbf{a})}\left[\nabla_\phi g_\phi(\mathbf{s}, \mathbf{a})\right] \\
&- \mathbb{E}_{\mu_\theta^t(\mathbf{s}, \mathbf{a})}\left[\frac{p_\theta^t(\mathbf{s})\exp(f_\phi(\mathbf{s}, \mathbf{a}))}{p_\theta^t(\mathbf{s})Z_\phi(\mathbf{s}, \mathbf{a})}\nabla_\phi f_\phi(\mathbf{s}, \mathbf{a}) + \lambda \frac{p_\theta^t(\mathbf{s})\exp(g_\phi(\mathbf{s}, \mathbf{a}))}{p_\theta^t(\mathbf{s})Z_\phi(\mathbf{s}, \mathbf{a})}\nabla_\phi g_\phi(\mathbf{s}, \mathbf{a})\right] \Big\},
\end{aligned}
\tag{27}
$$

By comparing Eq.(23) and Eq.(27), we can see that $\nabla_\phi \mathcal{L}_\lambda(\phi, \theta)$ and $\nabla_\phi \mathcal{M}_\lambda(\phi)$ have the same form and they only differs in the importance weights in the third and fourth terms. The importance weights used in $\nabla_\phi \mathcal{M}_\lambda(\phi)$ are correct and performing gradient ascent with $\nabla_\phi \mathcal{M}_\lambda(\phi)$ leads to a local maxima of $\mathcal{M}_\lambda(\phi)$ (assuming exact computation of the expectations). On the other hand, the importance weights used in $\nabla_\phi \mathcal{L}_\lambda(\phi, \theta)$ lead to bias in terms of the objective $\mathcal{M}_\lambda(\phi)$, and performing gradient ascent with $\nabla_\phi \mathcal{L}_\lambda(\phi, \theta)$ does not lead to a local maxima of $\mathcal{M}_\lambda(\phi)$. Thus, the global maximizer of $\mathcal{L}_\lambda(\phi, \theta)$ does not recover the true reward functions regardless of $\theta$.

The bias can be corrected only when $\lambda = 0$ and $\pi_\theta$ is the optimal maximum entropy policy of a reward $f_\phi(\mathbf{s}, \mathbf{a})$. This is because when $\lambda = 0$, $\mathcal{M}_0(\phi)$ is the maximum entropy IRL objective for learning the expert reward function and $\mathcal{L}_0(\phi, \theta)$ us the binary classification objective of GAIL and AIRL. As shown by Fu et al. (2018), the gradients of these two objectives are equivalent under the assumption that $\pi_\theta$ is the optimal maximum entropy policy for reward $r(\mathbf{s}, \mathbf{a}) = f_\phi(\mathbf{s}, \mathbf{a})$.

We emphasize that for our case it is insufficient to only assume that $\pi_\theta$ is the optimal maximum entropy policy without assuming that $\lambda = 0$. This is because to make the nominators equal, we need that the two equalities, $p_\theta^t(\mathbf{s})\exp(f_\phi(\mathbf{s}, \mathbf{a})) = p_f^t(\mathbf{s}, \mathbf{a})$ and $p_\theta^t(\mathbf{s})\exp(g_\phi(\mathbf{s}, \mathbf{a})) = p_g^t(\mathbf{s}, \mathbf{a})$, hold. The former holds when $\pi_\theta$ is (maximum entropy) optimal for a reward $f_\phi(\mathbf{s}, \mathbf{a})$, while the latter holds when $\pi_\theta$ is (maximum entropy) optimal for a reward $g_\phi(\mathbf{s}, \mathbf{a})$. Both cannot hold at the same time unless $f_\phi(\mathbf{s}, \mathbf{a}) = \alpha g_\phi(\mathbf{s}, \mathbf{a})$ with a positive scaling $\alpha$, which implies non-expert is expert himself. Moreover, to have that the partition functions are equivalent, $p_\theta^t(\mathbf{s})Z_\phi(\mathbf{s}, \mathbf{a}) = \mu_\theta^t(\mathbf{s}, \mathbf{a})$, we also need the policy $\pi_\theta$ to be optimal for a reward mixture. These contradictions imply that the two gradients can be made identical only when $\lambda = 0$.

## C    PSEUDOCODE

---

**Algorithm 1** Multiclass Generative Adversarial Imitation Learning (M-GAIL)

---

1: **Input:**  Demonstration datasets $\mathcal{D}_E$ and $\mathcal{D}_N$, initial parameters $\boldsymbol{\theta}$ and $\phi$, weight parameter $\lambda$.
2: **while** not converge **do**
3:     Collect trajectory samples $\{(\mathbf{s}_k, \mathbf{a}_k)\}_{k=1}^{K}$ using $\pi_{\boldsymbol{\theta}}(\mathbf{a}|\mathbf{s})$.
4:     Sample mini-batches data from $\mathcal{D}_E$ and $\mathcal{D}_N$.
5:     Update $\phi$ to maximize $\mathcal{L}_{\lambda}(\phi, \boldsymbol{\theta})$ in Eq.(5), by e.g., Adam (Kingma & Ba, 2014).
6:     Update $\boldsymbol{\theta}$ to solve the RL problem in Eq.(8), by e.g., TRPO (Schulman et al., 2015).
7: **end while**
8: **Output:** Parameterized policy $\pi_{\boldsymbol{\theta}}$.

---

## D    IMPLEMENTATION DETAILS

We implemented all methods with PyTorch deep learning framework (Paszke et al., 2017) (Our code will be publicly available.). All methods use a common setting as follows. The discriminator and policy are represented by neural networks with two hidden layers and 100 hyperbolic tangent units in each layer, as proposed by the original GAIL paper (Ho & Ermon, 2016). We use a Gaussian policy with state-independent and diagonal covariance for all tasks. The network parameters are initialized randomly. The discriminator is optimized by Adam (Kingma & Ba, 2014) with step-size $3 \times 10^{-4}$, $\beta_1 = 0.9$, and $\beta_2 = 0.999$. In each update iteration, we sample mini-batch of size 64 from $\mathcal{D}_E$ and $\mathcal{D}_N$ to update the discriminator.

We tried to make the implementation of GAIL, $\mathcal{U}$-GAIL, $\mathcal{V}$-GAIL, and M-GAIL as close as possible. In particular, we let $f_{\phi}(\mathbf{s}, \mathbf{a}) = 0$ be a constant function in $\mathcal{U}$-GAIL, $\mathcal{V}$-GAIL, and M-GAIL (This makes $\exp(f_{\phi}(\mathbf{s}, \mathbf{a})) = 1$). That is, the discriminator networks give two outputs; $g_{\phi}$ and $h_{\phi}$. This choice is valid since using three estimators to estimate three class posteriors is over-parameterize and we only need two estimators to compute an estimate of the third class posterior. With this implementation choice, $\mathcal{U}$-GAIL, $\mathcal{V}$-GAIL, M-GAIL with $\lambda = 0$ are exactly the same as GAIL where $h_{\phi}$ is the same as $d_{\phi}$ in GAIL.

The policy is optimized by trust-region policy optimization (TRPO) with generalized advantage estimation (GAE) (Schulman et al., 2016) with KL bound $\epsilon = 0.01$ and damping coefficient for the Fisher information matrix 0.1. We do not add causal entropy and set $\beta = 0$ for all methods. For GAE, we use the above network architecture to learn the value function with $\gamma = 0.995$ and $\lambda_{\mathrm{GAE}} = 0.97$. In each update iteration, we use Adam, with step-size $3 \times 10^{-4}$, $\beta_1 = 0.9$, $\beta_2 = 0.999$, and $\ell_2$ regularizer of $10^{-3}$, to update the value function network for 3 epochs with minibatch size 128. The GAE value is standardized to have zero mean and unit variance for TRPO update. In each update iteration, we collect transition samples of size $K$ using the current policy. This hyper-parameter setting is chosen for TRPO since it works well for learning the expert policy, as described below.

We consider four tasks; Half-Cheetah, Hopper, Walker2D, and Ant, simulated by PyBullet physics simulator (Coumans & Bai, 2018). (At the time of submission, OpenAI gym with Pybullet physics does not have the Swimmer task, and we could not successfully train a Humanoid expert agent. The two inverted-pendulum tasks are too simple with all methods perform equally well.) For these tasks, a trajectory (episode) consists of 1000 transition samples. To train the expert policy, we use TRPO with GAE with the above setting with $K = 2000$, except for Walker2D where we require an entropy regularizer of $\beta = 0.0001$ to learn a walking policy. Without this regularizer, the agent simply stands still to receive a reward of 1 in each time-step. The same issue does not happen when we train the IL agents and we do not use the entropy regularizer for IL. For each task, we train a TRPO agent for 5 trials with different random seeds and choose the best policy among 5 trials that gives the highest return at the 10000-th iteration as expert. The return of the chosen TRPO agent, as well as the returns at approximately $50\%$ and $70\%$ of the expert's return are shown in Figure 3. We can clearly see that non-expert policies can be learned using much less samples when compared to the expert policy.

After obtaining the expert and non-expert policies, we use the learned Gaussian policies to collect demonstration trajectories with exploration noise. The same datasets are used in all 5 trials of IL.

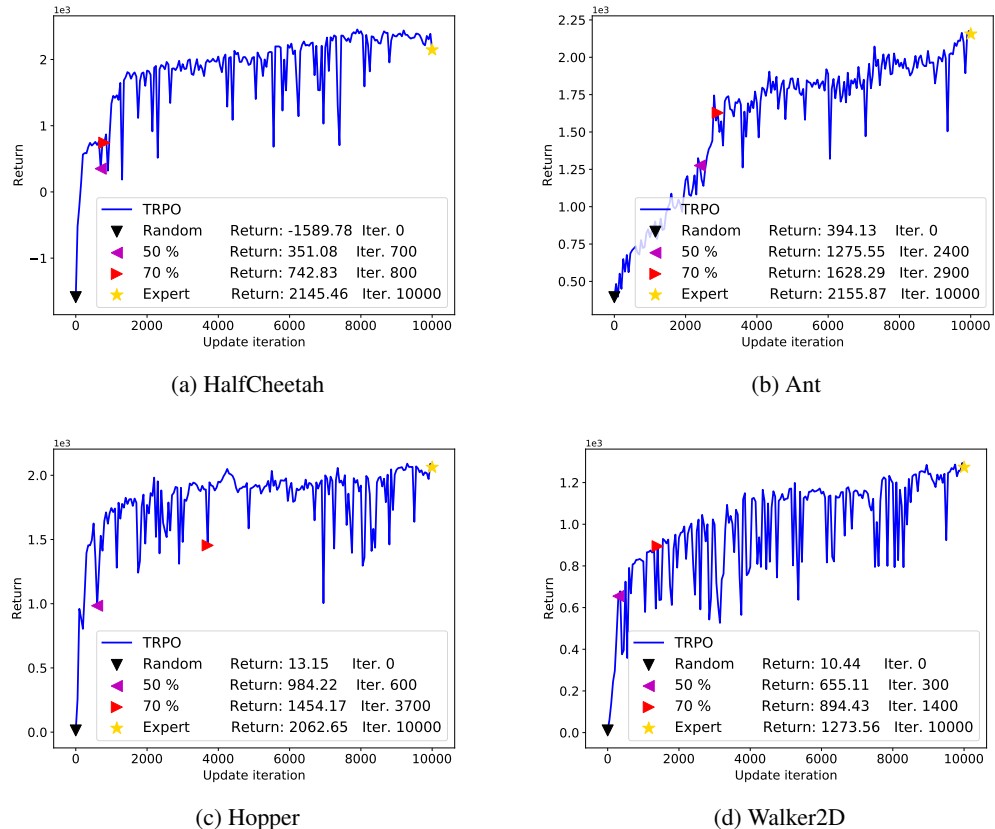

Figure 3: The return in each update iteration of TRPO expert agent. The return is computed using Gaussian policy with exploration noise. "Random" is the initial random policy. "Iter." denotes the update iteration to achieve the corresponding return values.

To sample expert demonstrations of size $N \in \{100, 300, 500\}$, which is smaller than a trajectory length, we use sub-trajectory sampling procedure as described by Ho & Ermon (2016). That is, we sample $2, 6, 10$ sub-trajectories of length 50 to obtain expert demonstrations of size $100, 300, 500$, respectively. For expert demonstrations of size $N \in \{1000, 10000\}$, we sample 1 and 10 full-length trajectories.

## E   MORE EXPERIMENTAL RESULTS

We train IL agents for 5 trials with different random seeds. All trials use the same datasets collected by the TRPO agent as explained previously. We evaluate the learned policies by generating 10 test trajectories using the learned policies to select actions without exploration noise. Then, we compute the averaged return over 10 trajectories using the true reward function without discount factor, and this gives us a learning curve for each trial.

Figures 4 to 7 show the results on each task for $N \in \{100, 300, 500\}$ and different values of $\lambda$, and Figures 8 to 11 show those for $N \in \{1000, 10000\}$. Tables 1 and 2 report the total return and final return, respectively, for $N \in \{100, 300, 500\}$, and Tables 3 and 4 report those for $N \in \{1000, 10000\}$. Based on Table 1, we choose the best overall value of $\lambda$ as follows; $\lambda = 0.5$ for M-GAIL, $\lambda = 0.01$ for $\mathcal{U}$-GAIL, and $\lambda = 0.01$ for $\mathcal{V}$-GAIL. We did not try different value of $\lambda$ and $\omega$ for MM-GAIL. Instead, we use $\lambda = 0.5$ and $\omega = 0.01$ since $\lambda = 0.5$ works well for multiclass classification in M-GAIL and the mixture coefficient of $0.01$ works well in $\mathcal{U}$-GAIL.

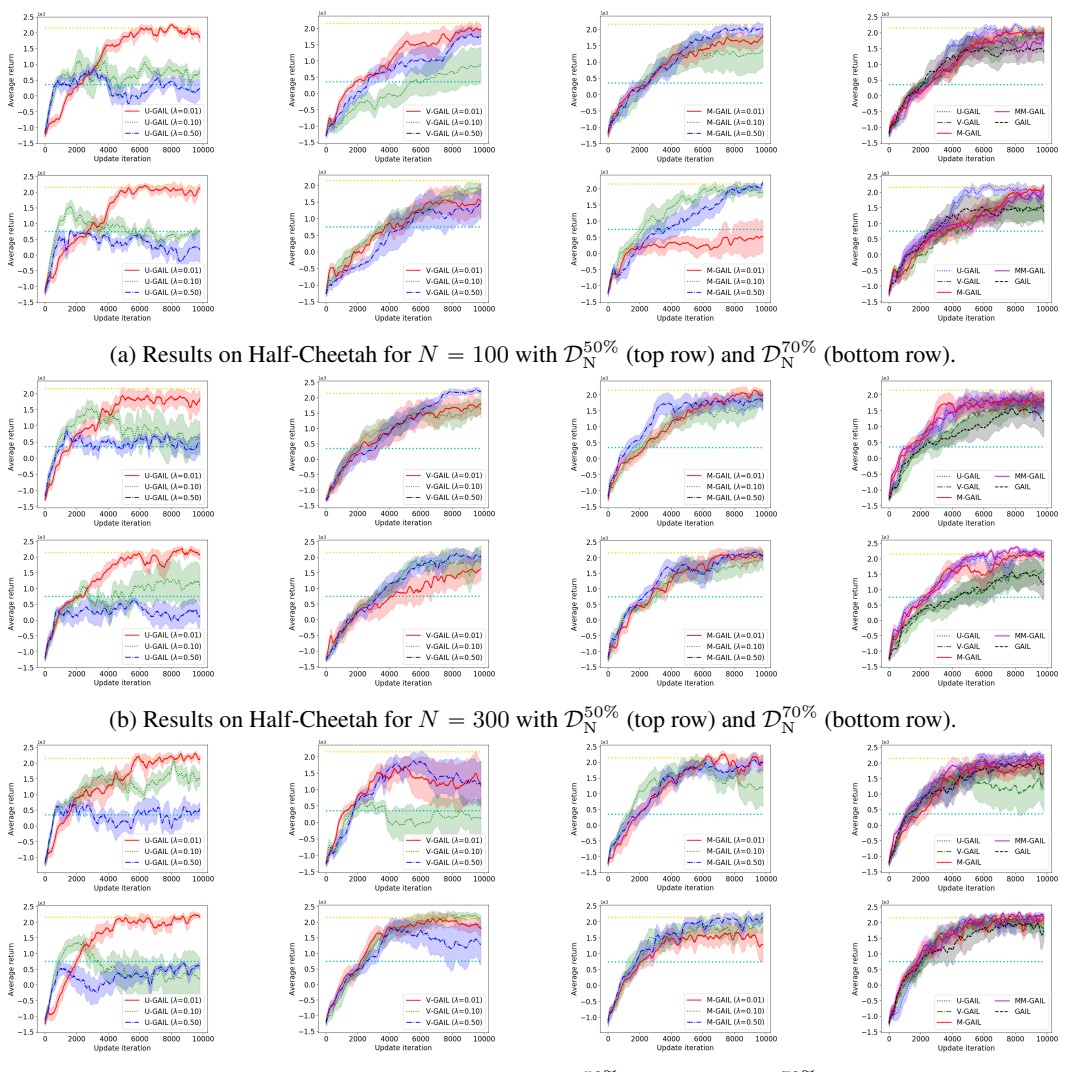

(a) Results on Half-Cheetah for $N = 100$ with $\mathcal{D}_{\mathrm{N}}^{50\%}$ (top row) and $\mathcal{D}_{\mathrm{N}}^{70\%}$ (bottom row).

(b) Results on Half-Cheetah for $N = 300$ with $\mathcal{D}_{\mathrm{N}}^{50\%}$ (top row) and $\mathcal{D}_{\mathrm{N}}^{70\%}$ (bottom row).

(c) Results on Half-Cheetah for $N = 500$ with $\mathcal{D}_{\mathrm{N}}^{50\%}$ (top row) and $\mathcal{D}_{\mathrm{N}}^{70\%}$ (bottom row).

Figure 4: Results on Half-Cheetah for $N \in \{100, 300, 500\}$ and $K = 1000$ on $\lambda \in \{0.01, 0.1, 0.5\}$. The right-most column shows the results of $\mathcal{U}$-GAIL for $\lambda = 0.01$, $\mathcal{V}$-GAIL for $\lambda = 0.01$, M-GAIL for $\lambda = 0.5$, and MM-GAIL for $\lambda = 0.5$ and $\omega = 0.01$.

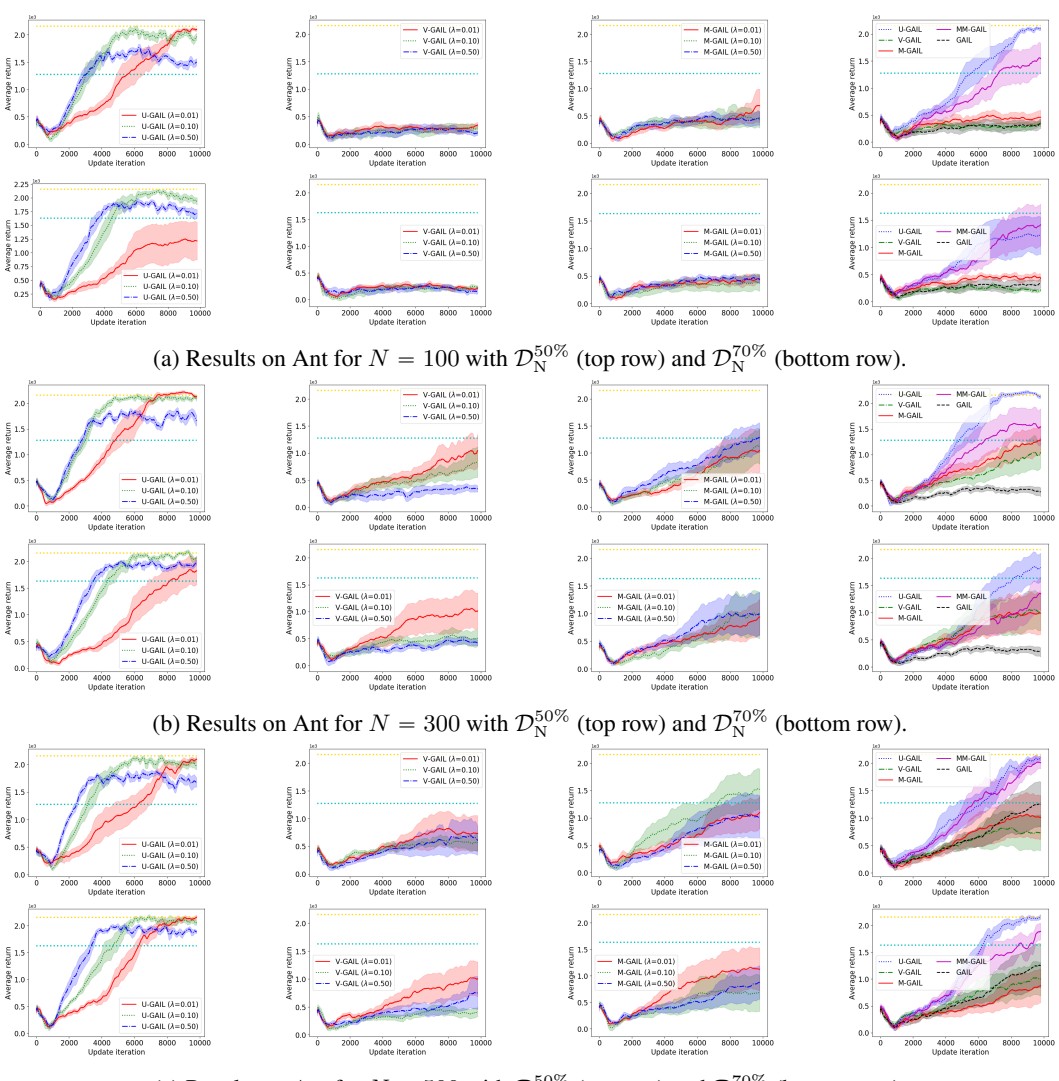

(a) Results on Ant for $N = 100$ with $\mathcal{D}_N^{50\%}$ (top row) and $\mathcal{D}_N^{70\%}$ (bottom row).

(b) Results on Ant for $N = 300$ with $\mathcal{D}_N^{50\%}$ (top row) and $\mathcal{D}_N^{70\%}$ (bottom row).

(c) Results on Ant for $N = 500$ with $\mathcal{D}_N^{50\%}$ (top row) and $\mathcal{D}_N^{70\%}$ (bottom row).

Figure 5: Results on Ant for $N \in \{100, 300, 500\}$ and $K = 1000$ on $\lambda \in \{0.01, 0.1, 0.5\}$. The right-most column shows the results of $\mathcal{U}$-GAIL for $\lambda = 0.01$, $\mathcal{V}$-GAIL for $\lambda = 0.01$, M-GAIL for $\lambda = 0.5$, and MM-GAIL for $\lambda = 0.5$ and $\omega = 0.01$.

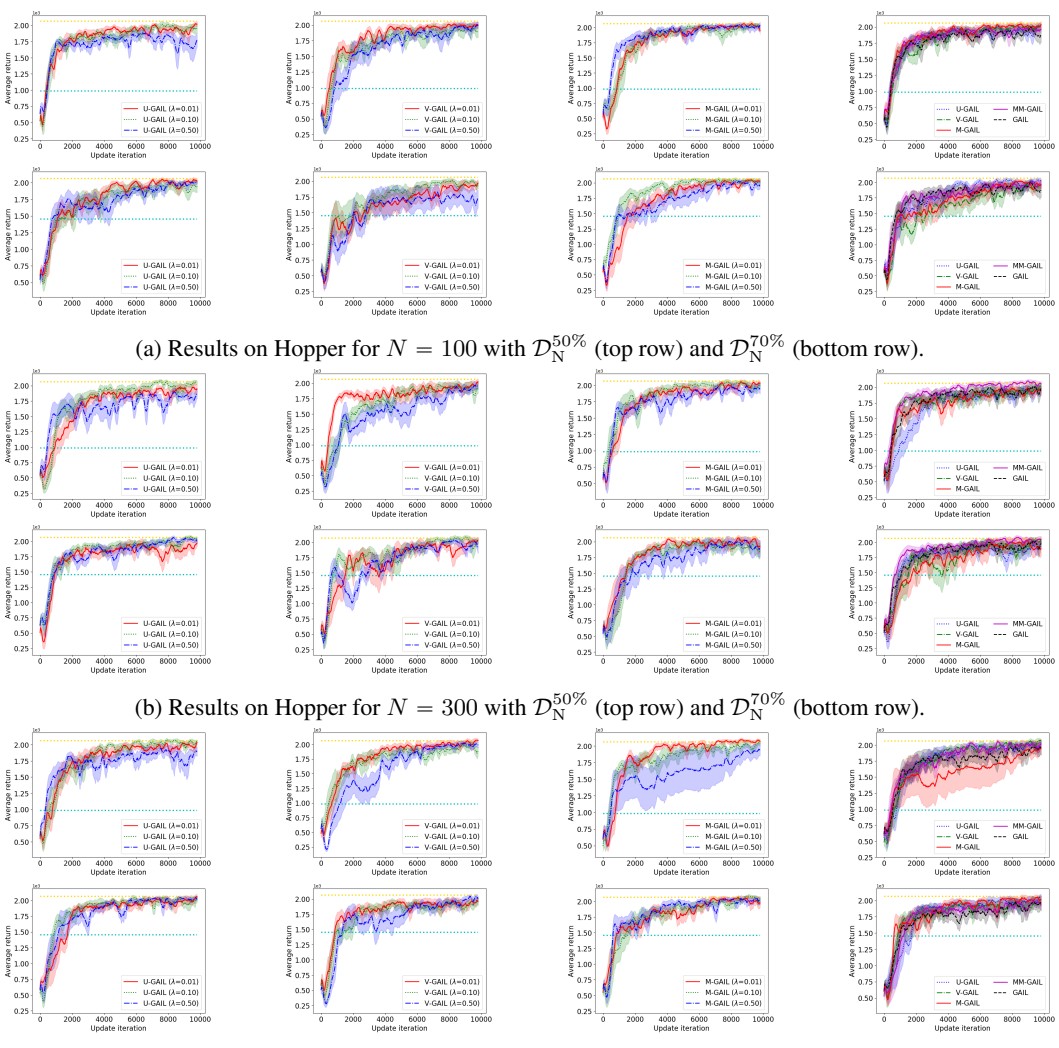

(a) Results on Hopper for $N = 100$ with $\mathcal{D}_N^{50\%}$ (top row) and $\mathcal{D}_N^{70\%}$ (bottom row).

(b) Results on Hopper for $N = 300$ with $\mathcal{D}_N^{50\%}$ (top row) and $\mathcal{D}_N^{70\%}$ (bottom row).

(c) Results on Hopper for $N = 500$ with $\mathcal{D}_N^{50\%}$ (top row) and $\mathcal{D}_N^{70\%}$ (bottom row).

Figure 6: Results on Hopper for $N \in \{100, 300, 500\}$ and $K = 1000$ on $\lambda \in \{0.01, 0.1, 0.5\}$. The right-most column shows the results of $\mathcal{U}$-GAIL for $\lambda = 0.01$, $\mathcal{V}$-GAIL for $\lambda = 0.01$, M-GAIL for $\lambda = 0.5$, and MM-GAIL for $\lambda = 0.5$ and $\omega = 0.01$.

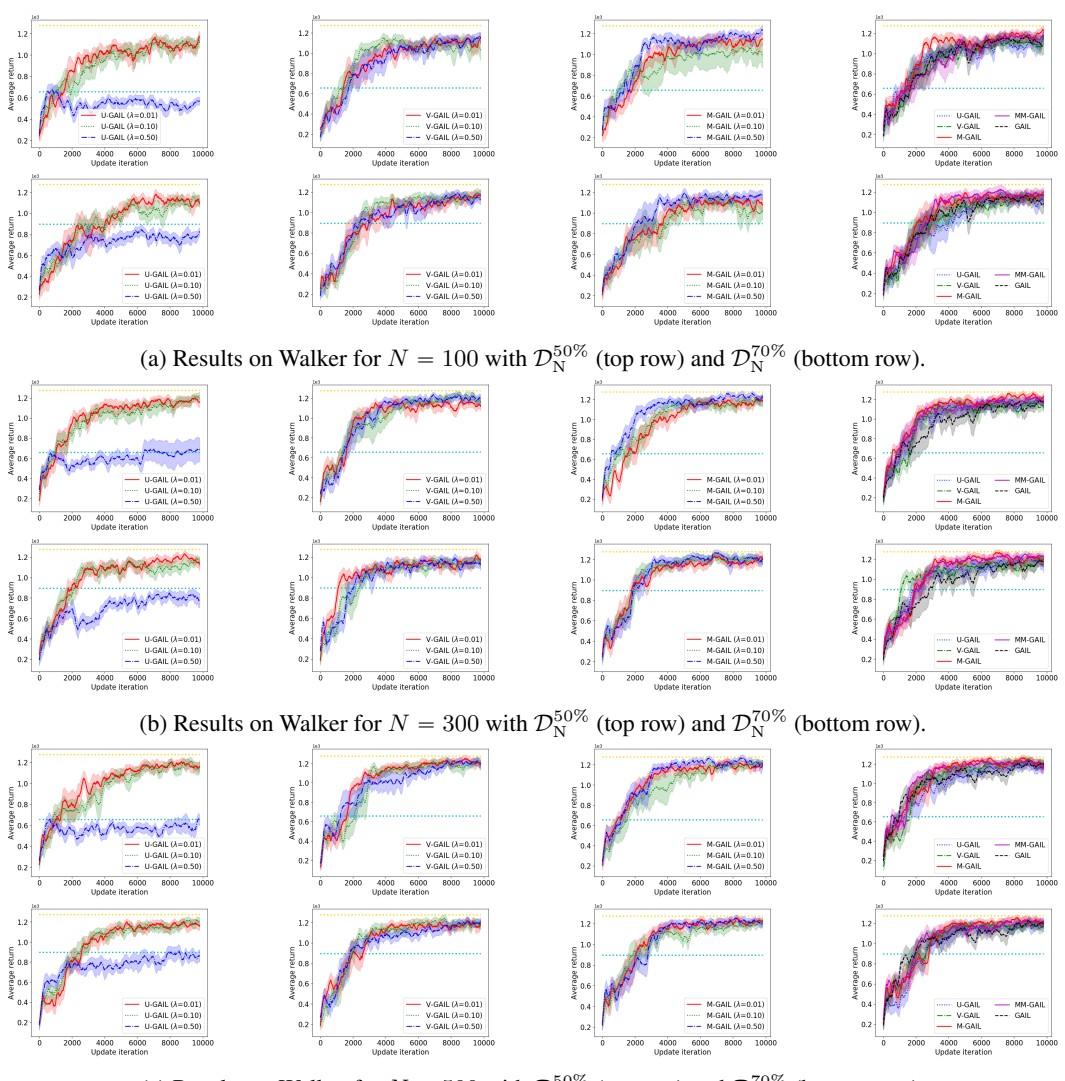

(a) Results on Walker for $N = 100$ with $\mathcal{D}_N^{50\%}$ (top row) and $\mathcal{D}_N^{70\%}$ (bottom row).

(b) Results on Walker for $N = 300$ with $\mathcal{D}_N^{50\%}$ (top row) and $\mathcal{D}_N^{70\%}$ (bottom row).

(c) Results on Walker for $N = 500$ with $\mathcal{D}_N^{50\%}$ (top row) and $\mathcal{D}_N^{70\%}$ (bottom row).

Figure 7: Results on Walker for $N \in \{100, 300, 500\}$ and $K = 1000$ on $\lambda \in \{0.01, 0.1, 0.5\}$. The right-most column shows the results of $\mathcal{U}$-GAIL for $\lambda = 0.01$, $\mathcal{V}$-GAIL for $\lambda = 0.01$, M-GAIL for $\lambda = 0.5$, and MM-GAIL for $\lambda = 0.5$ and $\omega = 0.01$.

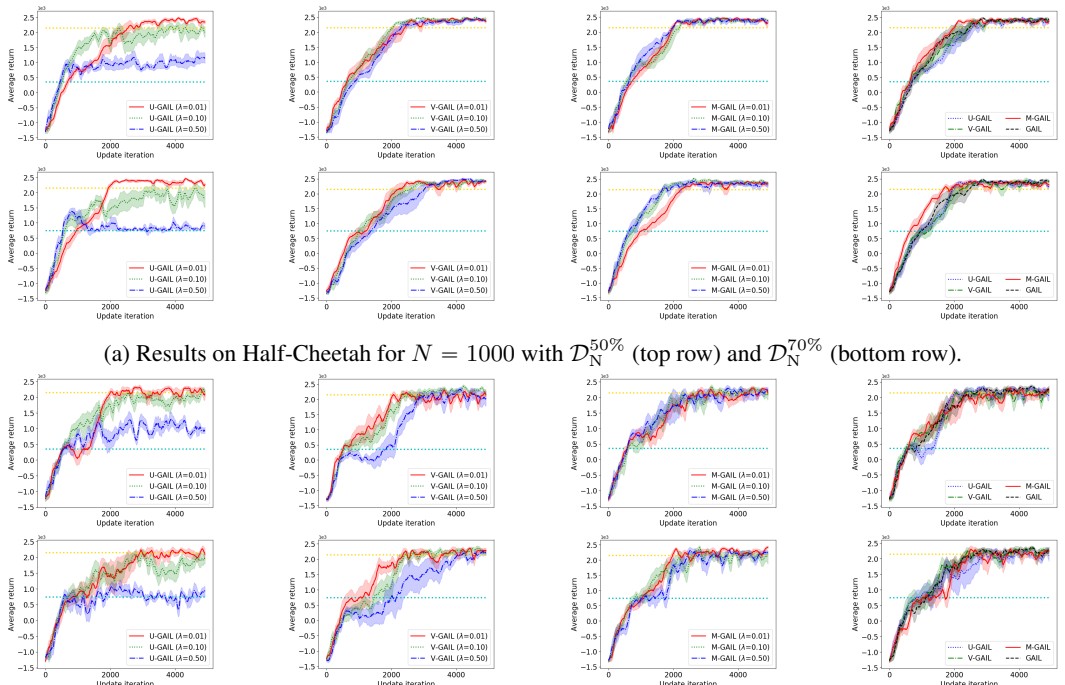

(a) Results on Half-Cheetah for $N = 1000$ with $\mathcal{D}_N^{50\%}$ (top row) and $\mathcal{D}_N^{70\%}$ (bottom row).

(b) Results on Half-Cheetah for $N = 10000$ with $\mathcal{D}_N^{50\%}$ (top row) and $\mathcal{D}_N^{70\%}$ (bottom row).

Figure 8: Results on Half-Cheetah for $N \in \{1000, 10000\}$ and $K = 2000$ on $\lambda \in \{0.01, 0.1, 0.5\}$. The right-most column shows the results of $\mathcal{U}$-GAIL for $\lambda = 0.01$, $\mathcal{V}$-GAIL for $\lambda = 0.01$, M-GAIL for $\lambda = 0.5$, and MM-GAIL for $\lambda = 0.5$ and $\omega = 0.01$.

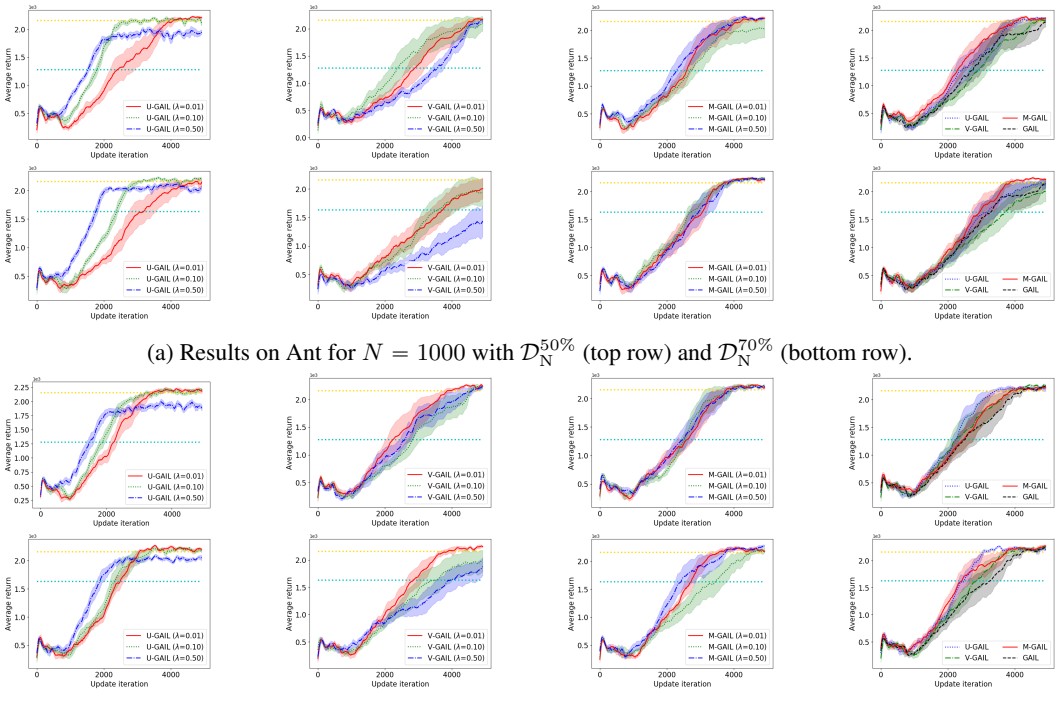

(a) Results on Ant for $N = 1000$ with $\mathcal{D}_N^{50\%}$ (top row) and $\mathcal{D}_N^{70\%}$ (bottom row).

(b) Results on Ant for $N = 10000$ with $\mathcal{D}_N^{50\%}$ (top row) and $\mathcal{D}_N^{70\%}$ (bottom row).

Figure 9: Results on Ant for $N \in \{1000, 10000\}$ and $K = 2000$ on $\lambda \in \{0.01, 0.1, 0.5\}$. The right-most column shows the results of $\mathcal{U}$-GAIL for $\lambda = 0.01$, $\mathcal{V}$-GAIL for $\lambda = 0.01$, M-GAIL for $\lambda = 0.5$, and MM-GAIL for $\lambda = 0.5$ and $\omega = 0.01$.

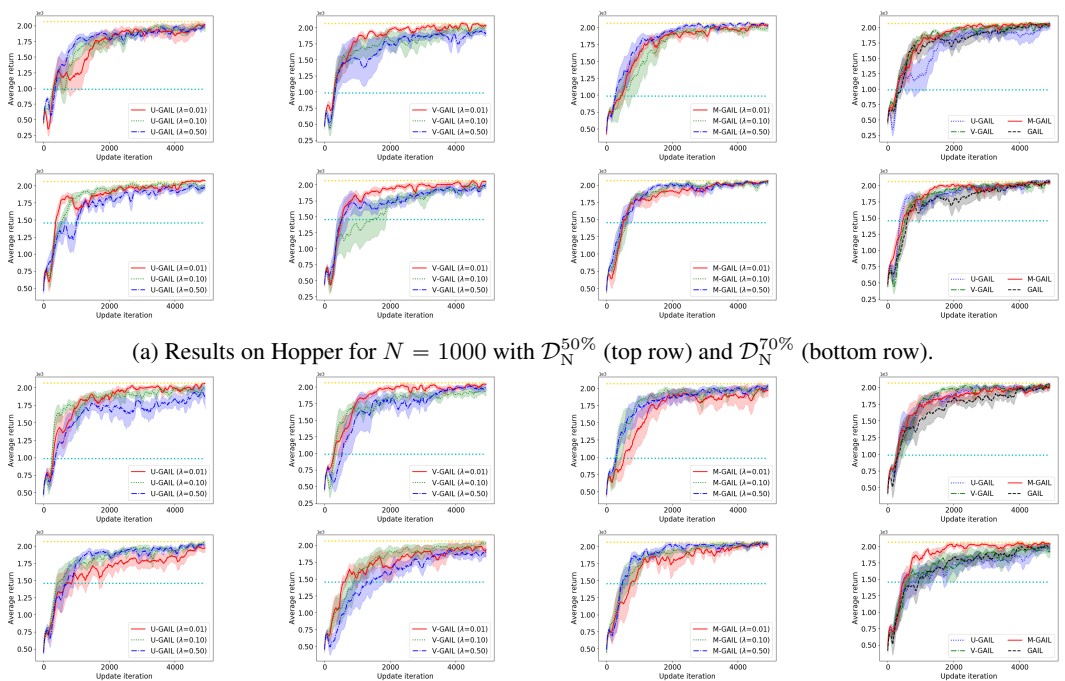

(a) Results on Hopper for $N = 1000$ with $\mathcal{D}_N^{50\%}$ (top row) and $\mathcal{D}_N^{70\%}$ (bottom row).

(b) Results on Hopper for $N = 10000$ with $\mathcal{D}_N^{50\%}$ (top row) and $\mathcal{D}_N^{70\%}$ (bottom row).

Figure 10: Results on Hopper for $N \in \{1000, 10000\}$ and $K = 2000$ on $\lambda \in \{0.01, 0.1, 0.5\}$. The right-most column shows the results of $\mathcal{U}$-GAIL for $\lambda = 0.01$, $\mathcal{V}$-GAIL for $\lambda = 0.01$, M-GAIL for $\lambda = 0.5$, and MM-GAIL for $\lambda = 0.5$ and $\omega = 0.01$.

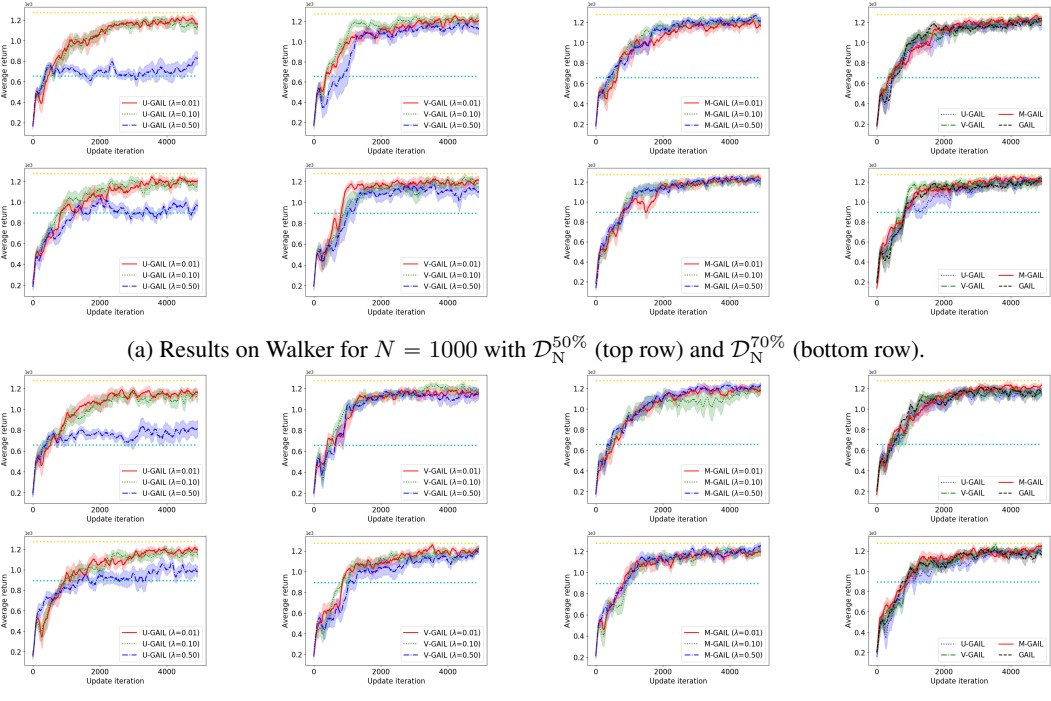

(a) Results on Walker for $N = 1000$ with $\mathcal{D}_N^{50\%}$ (top row) and $\mathcal{D}_N^{70\%}$ (bottom row).

(b) Results on Walker for $N = 10000$ with $\mathcal{D}_N^{50\%}$ (top row) and $\mathcal{D}_N^{70\%}$ (bottom row).

Figure 11: Results on Walker for $N \in \{1000, 10000\}$ and $K = 2000$ on $\lambda \in \{0.01, 0.1, 0.5\}$. The right-most column shows the results of $\mathcal{U}$-GAIL for $\lambda = 0.01$, $\mathcal{V}$-GAIL for $\lambda = 0.01$, M-GAIL for $\lambda = 0.5$, and MM-GAIL for $\lambda = 0.5$ and $\omega = 0.01$.

Table 1: The mean and standard error of total returns over 5 trials for $N \in \{100, 300, 500\}$ and $K = 1000$. The total returns are computed as the area under learning curves over 10000 update iteration divided by 10000. The best two methods in terms of the largest mean value are in boldface. For MM-GAIL, M-GAIL and $\mathcal{U}$-GAIL are used as $\lambda = 0.5$ and $\omega = 0.01$.

| Tasks | $\mathcal{D}_N^\%$ | N | $\mathcal{U}$-GAIL | | | $\mathcal{V}$-GAIL | | | M-GAIL | | | MM-GAIL | GAIL |
|---|---|---|---|---|---|---|---|---|---|---|---|---|---|
| | | | $\lambda=0.01$ | $\lambda=0.1$ | $\lambda=0.5z$ | $\lambda=0.01$ | $\lambda=0.1$ | $\lambda=0.5$ | $\lambda=0.01$ | $\lambda=0.1$ | $\lambda=0.5$ | | |
| Cheetah | 50 | 100 | **1229.1 (37.0)** | 604.2 (171.8) | 254.4 (123.5) | 1013.9 (160.1) | 58.5 (349.6) | 688.2 (232.8) | 931.6 (164.5) | 763.9 (383.7) | **1069.0 (143.7)** | 956.7 (235.9) | 922.3 (322.7) |
| | | 300 | 1212.1 (145.9) | 784.0 (286.6) | 375.1 (101.9) | 895.1 (349.0) | 915.7 (212.4) | 1028.3 (195.9) | 1090.1 (156.8) | 942.0 (231.6) | **1282.8 (206.8)** | **1238.0 (198.1)** | 726.5 (282.8) |
| | | 500 | **1343.4 (112.3)** | 1174.6 (75.0) | 286.7 (192.5) | 989.2 (337.4) | 82.6 (260.6) | 1002.6 (246.2) | 1212.9 (92.1) | 1187.0 (215.3) | 1208.1 (199.8) | **1449.3 (193.7)** | 1231.7 (280.6) |
| | 70 | 100 | **1310.9 (102.8)** | 737.3 (78.5) | 299.1 (121.9) | 731.1 (243.8) | 866.0 (109.6) | 470.6 (288.3) | 175.7 (233.2) | **1273.3 (128.7)** | 944.8 (198.2) | 919.1 (194.9) | 922.3 (322.7) |
| | | 300 | **1397.2 (57.5)** | 728.0 (318.2) | 195.7 (213.1) | 711.7 (333.8) | 1035.9 (169.6) | 1057.3 (198.2) | 1193.0 (212.1) | 1108.5 (231.0) | 1299.1 (157.4) | **1437.7 (61.4)** | 726.5 (282.8) |
| | | 500 | 1400.4 (78.2) | 469.4 (342.5) | 231.2 (144.0) | 1370.1 (181.4) | 1359.2 (149.2) | 1021.4 (301.7) | 1076.6 (210.7) | 1314.4 (220.1) | **1418.1 (147.1)** | **1485.4 (40.6)** | 1231.7 (280.6) |
| Ant | 50 | 100 | 1092.3 (96.8) | **1448.0 (53.0)** | **1260.1 (16.4)** | 275.5 (54.0) | 238.6 (66.7) | 230.1 (59.9) | 347.5 (81.3) | 367.3 (82.1) | 365.8 (70.5) | 781.0 (146.3) | 255.6 (71.6) |
| | | 300 | 1228.9 (67.9) | **1562.8 (24.3)** | **1371.8 (4.7)** | 559.8 (133.6) | 457.6 (103.6) | 263.4 (59.2) | 533.0 (150.4) | 545.2 (119.5) | 692.3 (114.5) | 908.7 (198.5) | 257.7 (32.4) |
| | | 500 | 1076.9 (125.4) | **1477.2 (41.8)** | **1433.9 (23.5)** | 528.0 (144.1) | 455.5 (60.4) | 430.1 (133.1) | 626.5 (133.7) | 870.7 (203.4) | 605.8 (187.8) | 1017.4 (110.8) | 617.5 (160.5) |
| | 70 | 100 | 743.5 (175.5) | **1395.5 (50.4)** | **1419.6 (32.7)** | 222.7 (28.3) | 186.0 (43.1) | 194.2 (42.6) | 338.4 (46.9) | 299.2 (90.3) | 363.6 (40.8) | 731.3 (159.9) | 255.6 (71.6) |
| | | 300 | 856.8 (129.2) | **1430.2 (36.4)** | **1479.4 (24.8)** | 648.1 (201.8) | 400.8 (70.5) | 323.0 (36.4) | 515.8 (117.4) | 506.3 (139.0) | 618.5 (173.2) | 613.8 (113.7) | 257.7 (32.4) |
| | | 500 | 1138.6 (55.8) | **1424.9 (33.3)** | **1498.9 (35.4)** | 604.4 (144.6) | 326.8 (42.3) | 412.6 (83.6) | 696.6 (212.6) | 456.2 (168.3) | 480.2 (130.4) | 940.7 (129.2) | 617.5 (160.5) |
| Hopper | 50 | 100 | 1791.2 (23.7) | 1779.1 (10.6) | 1684.2 (23.0) | 1773.8 (36.7) | 1680.2 (70.2) | 1611.1 (30.8) | 1760.5 (17.6) | 1758.2 (58.2) | **1841.0 (22.0)** | **1795.0 (22.1)** | 1753.3 (39.0) |
| | | 300 | 1663.5 (32.7) | 1750.1 (41.7) | 1630.3 (58.9) | 1771.0 (34.9) | 1635.7 (70.9) | 1505.4 (67.2) | 1770.2 (16.3) | **1833.4 (52.5)** | 1730.0 (18.7) | **1885.6 (31.3)** | 1768.7 (43.5) |
| | | 500 | 1754.5 (61.6) | 1777.9 (36.1) | 1702.0 (67.9) | 1766.6 (28.4) | 1717.0 (48.1) | 1561.0 (83.0) | **1834.1 (17.7)** | **1792.0 (50.3)** | 1563.4 (239.1) | 1772.7 (27.9) | 1706.9 (77.4) |
| | 70 | 100 | 1755.4 (43.7) | 1700.3 (88.6) | 1712.0 (72.6) | 1593.8 (86.0) | 1699.0 (65.9) | 1516.6 (125.4) | 1701.2 (59.3) | **1851.0 (18.8)** | 1673.3 (72.0) | **1757.2 (41.9)** | 1753.3 (39.0) |
| | | 300 | 1724.7 (41.5) | **1813.7 (39.3)** | 1775.9 (37.5) | 1681.0 (76.7) | 1756.2 (52.4) | 1654.0 (20.8) | 1793.7 (34.2) | 1721.7 (44.5) | 1666.6 (124.0) | **1859.9 (26.6)** | 1768.7 (43.5) |
| | | 500 | 1766.4 (21.5) | **1816.7 (21.6)** | 1776.7 (47.1) | 1765.0 (49.4) | 1713.4 (14.5) | 1644.5 (68.4) | 1763.8 (32.7) | 1754.7 (58.6) | **1799.5 (32.9)** | 1734.9 (34.1) | 1706.9 (77.4) |
| Walker | 50 | 100 | 939.0 (31.9) | 918.5 (35.6) | 537.9 (13.6) | 923.9 (35.1) | **956.6 (29.1)** | 902.6 (44.1) | 940.1 (44.3) | 888.1 (101.5) | **1011.5 (19.2)** | 938.4 (39.9) | 929.7 (29.0) |
| | | 300 | 1003.4 (30.2) | 980.6 (30.4) | 596.5 (47.7) | 987.8 (30.0) | 989.5 (37.5) | 997.5 (8.8) | 949.5 (16.3) | 989.6 (33.2) | **1071.7 (19.5)** | **1034.4 (32.7)** | 946.1 (42.4) |
| | | 500 | 989.2 (21.7) | 945.5 (29.3) | 557.8 (14.2) | 1019.9 (18.4) | 984.4 (23.2) | 979.2 (38.7) | 1036.2 (11.1) | 974.9 (30.6) | **1058.4 (16.0)** | **1067.3 (5.9)** | 1009.9 (21.9) |
| | 70 | 100 | 909.1 (40.2) | 894.6 (21.6) | 717.4 (15.3) | 925.1 (44.8) | 951.9 (14.9) | 930.1 (38.8) | 915.5 (52.0) | 891.4 (51.5) | **987.6 (14.0)** | **994.9 (16.8)** | 929.7 (29.0) |
| | | 300 | 1001.1 (15.0) | 984.0 (16.5) | 685.0 (17.5) | 1033.9 (18.5) | 983.2 (16.1) | 985.8 (22.5) | 1019.5 (27.3) | **1045.2 (14.4)** | 1041.7 (16.0) | **1042.4 (17.7)** | 946.1 (42.4) |
| | | 500 | 971.7 (25.1) | 984.9 (15.2) | 762.6 (22.7) | 1015.0 (16.0) | 1021.1 (21.0) | 991.5 (17.5) | **1043.2 (20.5)** | 1008.1 (23.9) | 1029.8 (17.3) | **1038.5 (29.1)** | 1009.9 (21.9) |

Table 2: The mean and standard error of final average returns over 5 trials for $N \in \{100, 300, 500\}$ and $K = 1000$. The final average returns are computed as average returns obtained from the last 500 update iteration. The best two methods for each row in terms of the largest mean value are in boldface.

| Tasks | $\mathcal{D}_N^\%$ | $N$ | U-GAIL | | | V-GAIL | | | M-GAIL | | | MM-GAIL | GAIL |
|---|---|---|---|---|---|---|---|---|---|---|---|---|---|
| | | | $\lambda = 0.01$ | $\lambda = 0.1$ | $\lambda = 0.5z$ | $\lambda = 0.01$ | $\lambda = 0.1$ | $\lambda = 0.5$ | $\lambda = 0.01$ | $\lambda = 0.1$ | $\lambda = 0.5$ | | |
| Cheetah | 50% | 100 | 1931.2 (60.4) | 696.0 (78.0) | 177.2 (317.5) | **1976.1 (174.4)** | 846.5 (553.0) | 1740.5 (217.1) | 1705.4 (197.5) | 1303.2 (510.8) | **2001.4 (191.1)** | 1752.4 (265.1) | 1453.9 (366.2) |
| | | 300 | 1744.1 (298.4) | 636.5 (504.3) | 375.9 (304.6) | 1752.1 (302.2) | 1661.4 (268.7) | **2227.9 (83.7)** | **2028.7 (148.1)** | 1810.3 (245.5) | 1841.4 (243.4) | 1790.9 (242.9) | 1267.3 (487.8) |
| | | 500 | **2215.2 (63.4)** | 1508.3 (196.9) | 484.7 (167.6) | 1279.2 (661.0) | 143.7 (563.0) | 1396.2 (651.7) | 2000.6 (204.0) | 1175.8 (576.5) | 1953.0 (234.6) | **2099.9 (192.2)** | 1738.4 (365.5) |
| | 70% | 100 | **2005.7 (149.5)** | 713.6 (80.7) | 217.4 (425.3) | 1505.6 (459.5) | 1896.9 (177.5) | 1399.4 (444.9) | 502.6 (513.4) | 1890.7 (139.3) | **2087.1 (56.7)** | 1868.5 (114.3) | 1453.9 (366.2) |
| | | 300 | **2129.1 (83.1)** | 1161.9 (576.2) | 211.2 (321.7) | 1562.6 (361.2) | 1956.4 (371.2) | 2040.8 (193.9) | 2077.1 (203.8) | 1902.7 (304.1) | 2099.2 (94.3) | **2164.2 (57.5)** | 1267.3 (487.8) |
| | | 500 | **2201.5 (40.8)** | 260.7 (693.7) | 581.0 (60.4) | 1871.0 (192.5) | 2161.5 (86.3) | 1370.6 (601.1) | 1283.0 (451.5) | 1977.9 (295.5) | 2082.0 (151.9) | **2164.4 (56.7)** | 1738.4 (365.5) |
| Ant | 50% | 100 | **2097.7 (14.4)** | **1937.2 (45.3)** | 1495.6 (58.5) | 327.9 (45.9) | 281.1 (78.9) | 201.1 (39.7) | 667.4 (284.9) | 429.3 (144.0) | 450.7 (120.6) | 1525.7 (288.4) | 315.5 (85.1) |
| | | 300 | **2152.5 (19.8)** | **2087.1 (22.1)** | 1739.7 (86.5) | 1020.8 (301.4) | 820.9 (277.3) | 343.4 (59.6) | 1032.8 (396.6) | 1116.0 (310.9) | 1266.3 (265.7) | 1554.9 (336.4) | 280.9 (79.8) |
| | | 500 | **2084.1 (25.9)** | **2006.2 (60.7)** | 1673.5 (95.3) | 733.2 (319.6) | 560.9 (136.1) | 642.1 (284.1) | 1065.4 (287.8) | 1513.5 (370.0) | 1016.1 (398.3) | 2004.4 (123.1) | 1247.2 (407.0) |
| | 70% | 100 | 1211.5 (346.6) | **1960.1 (41.2)** | **1713.3 (70.0)** | 204.3 (15.4) | 239.9 (27.4) | 153.9 (38.1) | 438.0 (80.4) | 388.4 (146.5) | 446.6 (61.7) | 1385.3 (380.7) | 315.5 (85.1) |
| | | 300 | 1822.7 (261.6) | **2060.1 (31.5)** | **1938.8 (31.7)** | 1029.5 (340.1) | 499.0 (147.9) | 427.8 (54.0) | 899.9 (309.2) | 985.1 (416.5) | 979.7 (382.3) | 1316.3 (292.4) | 280.9 (79.8) |
| | | 500 | **2138.4 (26.7)** | **2078.6 (23.3)** | 1889.4 (29.9) | 1011.8 (307.2) | 397.0 (84.0) | 749.0 (282.8) | 1126.8 (394.2) | 669.4 (344.8) | 859.0 (315.6) | 1857.6 (150.1) | 1247.2 (407.0) |
| Hopper | 50% | 100 | 1999.5 (22.6) | 1961.5 (35.2) | 1694.5 (154.6) | 1994.1 (38.1) | 1913.4 (59.2) | 1982.4 (37.7) | 1997.6 (24.4) | **2030.0 (22.5)** | **2016.8 (45.0)** | 1957.4 (26.8) | 1867.4 (75.0) |
| | | 300 | 1948.3 (55.6) | **2048.1 (11.1)** | 1820.4 (75.5) | 1992.9 (32.8) | 1882.5 (80.0) | 1950.7 (52.6) | 2028.5 (10.0) | 1983.5 (31.4) | 1946.7 (50.0) | **2053.1 (17.4)** | 1961.1 (54.9) |
| | | 500 | 1984.8 (37.3) | 2034.8 (18.3) | 1861.3 (73.4) | **2042.1 (49.4)** | 1918.3 (49.7) | 1986.7 (33.6) | **2069.5 (15.8)** | 2025.8 (26.3) | 1926.1 (94.0) | 1992.1 (73.4) | 1950.0 (105.3) |
| | 70% | 100 | **2028.2 (15.5)** | 1946.1 (79.4) | 1999.8 (16.8) | 1944.9 (54.0) | 1961.4 (9.8) | 1705.6 (122.5) | **2023.7 (20.2)** | 2023.4 (44.7) | 1966.6 (42.4) | 1955.4 (45.4) | 1867.4 (75.0) |
| | | 300 | 1925.8 (50.5) | **2052.0 (12.9)** | 2015.3 (15.6) | 2020.3 (49.7) | 1959.5 (88.3) | 1960.6 (56.1) | 1969.5 (67.1) | 1882.5 (90.2) | 1902.3 (116.3) | **2038.3 (41.6)** | 1961.1 (54.9) |
| | | 500 | 2017.3 (16.0) | 1990.6 (37.2) | 2020.6 (30.9) | 1952.5 (27.7) | 1939.1 (53.3) | 2000.8 (23.3) | 1991.7 (63.7) | **2026.4 (42.8)** | **2028.2 (32.8)** | 1949.3 (31.9) | 1950.0 (105.3) |
| Walker | 50% | 100 | 1113.0 (24.5) | 1094.4 (43.1) | 561.8 (11.9) | 1103.3 (30.1) | 1060.7 (77.0) | 1143.2 (21.2) | 1118.6 (58.5) | 1007.2 (112.0) | **1213.3 (23.1)** | **1156.7 (18.0)** | 1096.4 (66.4) |
| | | 300 | 1177.8 (20.0) | **1213.0 (19.1)** | 678.8 (115.3) | 1122.1 (25.6) | 1169.9 (20.1) | 1194.1 (21.3) | 1189.6 (17.9) | 1185.8 (38.6) | **1212.4 (9.1)** | 1187.4 (16.5) | 1176.7 (11.6) |
| | | 500 | 1155.3 (16.4) | 1150.0 (25.9) | 609.3 (46.2) | 1203.5 (38.3) | 1205.9 (24.5) | **1209.4 (13.9)** | 1193.4 (8.5) | **1209.9 (6.8)** | 1201.2 (22.0) | 1188.6 (21.7) | 1197.1 (10.2) |
| | 70% | 100 | 1102.6 (31.6) | 1133.8 (19.2) | 796.6 (20.6) | 1166.0 (20.8) | **1176.8 (27.8)** | 1140.1 (33.6) | 1104.2 (39.2) | 1008.1 (75.8) | 1168.1 (25.9) | **1183.5 (33.7)** | 1096.4 (66.4) |
| | | 300 | 1150.1 (28.6) | 1154.5 (9.2) | 798.3 (46.0) | 1188.2 (21.6) | 1162.2 (14.9) | 1134.5 (21.1) | **1202.2 (33.6)** | 1195.2 (15.6) | 1194.9 (25.0) | **1230.8 (11.7)** | 1176.7 (11.6) |
| | | 500 | 1174.3 (22.3) | 1193.5 (17.6) | 857.8 (33.1) | 1168.5 (22.9) | 1189.8 (20.7) | 1196.5 (19.9) | **1225.3 (7.7)** | 1203.8 (12.9) | **1209.3 (12.6)** | 1208.2 (25.1) | 1197.1 (10.2) |

Table 3: The mean and standard error of total average returns over 5 trials for $N \in \{1000, 100000\}$ and $K = 2000$. The total average returns are computed as the area under learning curves over 10000 update iteration divided by 10000. The best two methods in terms of the largest mean value are in boldface.

| Tasks | $\mathcal{D}_N$ | $N$ | $\mathcal{U}$-GAIL $\lambda = 0.01$ | $\mathcal{U}$-GAIL $\lambda = 0.1$ | $\mathcal{U}$-GAIL $\lambda = 0.5$ | $\mathcal{V}$-GAIL $\lambda = 0.01$ | $\mathcal{V}$-GAIL $\lambda = 0.1$ | $\mathcal{V}$-GAIL $\lambda = 0.5$ | M-GAIL $\lambda = 0.01$ | M-GAIL $\lambda = 0.1$ | M-GAIL $\lambda = 0.5$ | GAIL |
|---|---|---|---|---|---|---|---|---|---|---|---|---|
| Cheetah | 50% | 1000 | 1517.0 (63.8) | 1556.3 (110.8) | 795.0 (82.2) | 1617.8 (91.5) | **1664.2 (39.6)** | 1495.8 (68.8) | 1625.9 (44.6) | 1642.8 (53.8) | **1724.3 (49.8)** | 1619.8 (87.5) |
| | | 10000 | 1458.9 (30.0) | 1432.9 (61.3) | 744.2 (52.9) | **1489.0 (73.2)** | 1461.1 (63.0) | 1081.3 (93.8) | 1417.2 (56.9) | 1442.7 (48.4) | 1476.7 (65.5) | **1499.6 (49.7)** |
| | 70% | 1000 | 1619.0 (54.6) | 1407.4 (129.0) | 734.8 (33.1) | 1572.9 (82.1) | 1511.1 (84.6) | 1375.0 (118.9) | 1551.8 (65.9) | **1788.5 (44.9)** | **1788.0 (36.9)** | 1619.8 (87.5) |
| | | 10000 | 1418.4 (161.6) | 1303.6 (99.7) | 605.7 (26.2) | **1534.3 (85.2)** | 1377.4 (78.4) | 990.5 (207.7) | **1572.5 (32.6)** | 1502.6 (82.7) | 1388.3 (45.1) | 1499.6 (49.7) |
| Ant | 50% | 1000 | 1241.0 (91.2) | **1537.1 (18.2)** | **1497.3 (22.3)** | 1116.0 (97.7) | 1236.7 (148.4) | 956.6 (45.8) | 1242.9 (86.1) | 1174.2 (112.1) | 1336.7 (86.4) | 1150.4 (128.5) |
| | | 10000 | 1383.7 (49.0) | **1489.9 (19.7)** | **1486.3 (32.3)** | 1315.6 (89.4) | 1123.6 (91.4) | 1204.7 (98.0) | 1281.4 (62.6) | 1288.4 (50.5) | 1301.4 (82.4) | 1232.8 (92.4) |
| | 70% | 1000 | 1189.8 (96.4) | 1440.5 (22.4) | **1576.8 (11.6)** | 1082.6 (123.2) | 1056.8 (118.7) | 738.2 (94.1) | 1293.4 (65.1) | 1345.4 (49.6) | 1274.4 (63.7) | 1150.4 (128.5) |
| | | 10000 | **1419.4 (21.0)** | 1438.7 (49.8) | **1510.7 (24.9)** | 1297.3 (61.8) | 1117.1 (116.9) | 1041.9 (97.7) | 1348.0 (38.4) | 1186.8 (90.0) | 1406.2 (74.5) | 1232.8 (92.4) |
| Hopper | 50% | 1000 | 1653.0 (99.1) | 1687.8 (72.4) | 1732.5 (34.1) | **1823.9 (19.3)** | 1713.7 (99.6) | 1654.0 (115.6) | 1746.3 (22.7) | 1702.7 (51.2) | **1822.7 (29.7)** | 1744.4 (65.7) |
| | | 10000 | **1793.9 (30.8)** | 1789.5 (32.8) | 1611.6 (83.7) | **1807.2 (36.8)** | 1711.9 (36.1) | 1655.2 (93.2) | 1656.8 (119.4) | 1787.4 (41.8) | 1777.4 (65.8) | 1679.2 (45.5) |
| | 70% | 1000 | 1825.4 (17.7) | 1829.4 (26.8) | 1704.8 (24.5) | 1808.5 (26.8) | 1616.6 (91.3) | 1698.2 (64.3) | 1805.7 (30.0) | **1836.1 (14.7)** | **1845.7 (28.5)** | 1744.4 (65.7) |
| | | 10000 | 1644.8 (84.0) | 1790.2 (34.2) | 1769.6 (27.9) | 1696.9 (91.1) | 1729.7 (77.7) | 1545.7 (88.1) | 1749.4 (58.3) | **1825.9 (31.5)** | **1851.6 (21.5)** | 1679.6 (45.5) |
| Walker | 50% | 1000 | 1022.4 (15.4) | 1018.5 (21.2) | 682.8 (25.1) | 1047.3 (14.3) | **1089.6 (7.1)** | 973.5 (34.5) | 1011.2 (27.4) | **1056.6 (11.7)** | 1047.0 (12.5) | 1037.0 (11.9) |
| | | 10000 | 1008.4 (20.3) | 985.7 (18.1) | 735.4 (19.6) | 1024.7 (8.4) | **1049.1 (16.1)** | 1005.8 (17.5) | 1016.2 (15.1) | 992.7 (41.6) | **1041.9 (15.6)** | 1032.5 (17.5) |
| | 70% | 1000 | 1018.1 (14.0) | 1044.4 (13.4) | 854.6 (8.6) | 1069.8 (20.0) | 1028.4 (31.5) | 982.3 (35.2) | 1049.2 (19.2) | **1068.0 (14.6)** | **1070.5 (8.3)** | 1037.0 (11.9) |
| | | 10000 | 1002.0 (20.4) | 993.4 (6.5) | 879.4 (24.7) | 1036.1 (11.2) | 1024.8 (4.9) | 962.9 (23.9) | **1045.7 (9.2)** | 1038.1 (14.8) | **1061.1 (13.5)** | 1032.5 (17.5) |

Table 4: The mean and standard error of final average returns over 5 trials for $N \in \{1000, 100000\}$ and $K = 2000$. The final average returns are computed as average returns obtained from the last 500 update iteration. The best two methods for each row in terms of the largest mean value are in boldface.

| Tasks | $\mathcal{D}_N$ | $N$ | $\mathcal{U}$-GAIL | | | $\mathcal{V}$-GAIL | | | M-GAIL | | | GAIL |
|---|---|---|---|---|---|---|---|---|---|---|---|---|
| | | | $\lambda = 0.01$ | $\lambda = 0.1$ | $\lambda = 0.5$ | $\lambda = 0.01$ | $\lambda = 0.1$ | $\lambda = 0.5$ | $\lambda = 0.01$ | $\lambda = 0.1$ | $\lambda = 0.5$ | |
| Cheetah | 50% | 1000 | 2383.6 (52.1) | 2041.6 (101.3) | 1107.6 (99.0) | 2404.9 (11.0) | **2451.8 (17.3)** | 2413.7 (21.6) | 2367.4 (24.5) | 2413.0 (10.0) | 2375.2 (47.4) | **2414.8 (32.8)** |
| | | 10000 | 2169.9 (43.9) | 2065.6 (50.7) | 953.2 (13.3) | 2102.8 (56.8) | **2296.1 (20.6)** | 2080.1 (203.4) | 2191.9 (32.0) | 2178.4 (64.4) | 2142.1 (58.8) | **2194.3 (26.1)** |
| | 70% | 1000 | 2330.8 (19.0) | 1968.5 (132.0) | 831.2 (46.8) | 2362.3 (25.1) | **2424.8 (22.4)** | **2418.1 (6.9)** | 2351.7 (37.8) | 2403.1 (17.6) | 2307.9 (41.2) | 2414.8 (32.8) |
| | | 10000 | 2107.6 (93.9) | 1903.5 (63.1) | 839.4 (145.4) | 2222.2 (46.6) | **2291.1 (29.8)** | 2160.5 (25.1) | **2294.3 (32.6)** | 2087.6 (115.0) | 2151.4 (22.3) | 2194.3 (26.1) |
| Ant | 50% | 1000 | **2207.0 (5.4)** | 2161.4 (8.8) | 1940.8 (46.9) | 2159.1 (26.3) | 2042.7 (170.4) | 2096.0 (33.1) | 2200.4 (21.8) | 2022.0 (176.5) | **2204.5 (13.4)** | 2051.1 (141.7) |
| | | 10000 | 2192.5 (25.1) | 2183.3 (28.5) | 1927.0 (17.4) | **2235.3 (5.0)** | 2191.8 (45.4) | 2182.8 (41.4) | **2220.5 (7.9)** | 2215.6 (12.5) | 2202.1 (4.6) | 2202.8 (13.5) |
| | 70% | 1000 | 2131.5 (52.1) | 2192.6 (7.9) | 2016.7 (30.1) | 1965.0 (192.0) | 1947.3 (238.9) | 1389.6 (253.5) | 2216.2 (14.9) | **2230.1 (2.2)** | **2225.9 (3.8)** | 2051.1 (141.7) |
| | | 10000 | 2190.2 (16.9) | 2182.3 (4.5) | 2028.3 (10.5) | **2224.5 (19.2)** | 1952.7 (209.9) | 1794.9 (168.5) | 2199.0 (24.7) | 2175.0 (41.3) | **2234.8 (22.1)** | 2202.8 (13.5) |
| Hopper | 50% | 1000 | 1985.6 (62.1) | 1981.4 (25.7) | 1955.8 (18.7) | **2042.4 (13.1)** | 1982.3 (41.6) | 1924.6 (33.4) | 2032.0 (8.2) | 1987.5 (20.5) | **2044.1 (14.3)** | 2035.1 (28.0) |
| | | 10000 | **2020.7 (18.1)** | 1964.7 (27.5) | 1896.8 (94.2) | **2021.5 (15.3)** | 1928.7 (50.3) | 1979.0 (33.2) | 1953.2 (99.6) | 1971.2 (31.8) | 2011.3 (19.0) | 1989.9 (24.8) |
| | 70% | 1000 | **2069.4 (4.3)** | 2001.1 (16.7) | 1971.3 (40.9) | 2027.6 (23.2) | 1971.5 (18.8) | 1946.4 (36.8) | **2039.8 (10.6)** | 2016.4 (13.0) | 2031.9 (13.3) | 2035.1 (28.0) |
| | | 10000 | 1962.3 (53.7) | 2027.2 (26.3) | 2005.8 (15.9) | 1950.4 (73.5) | 2002.1 (27.2) | 1872.6 (61.9) | 2023.0 (21.0) | **2050.8 (9.0)** | **2041.6 (7.7)** | 1989.9 (24.8) |
| Walker | 50% | 1000 | 1198.0 (13.1) | 1144.7 (19.6) | 768.8 (45.6) | 1202.7 (13.0) | 1222.1 (7.3) | 1147.2 (35.2) | 1178.9 (18.2) | **1219.4 (13.2)** | **1230.8 (14.0)** | 1207.1 (4.9) |
| | | 10000 | 1159.1 (15.2) | 1116.8 (34.8) | 806.6 (58.0) | 1152.5 (19.1) | 1185.8 (7.1) | 1135.7 (44.8) | **1185.6 (10.9)** | 1177.3 (20.6) | **1211.1 (12.8)** | 1155.0 (16.6) |
| | 70% | 1000 | 1200.5 (9.9) | 1176.1 (14.6) | 939.9 (26.8) | 1192.6 (22.3) | 1199.8 (9.0) | 1114.2 (27.1) | **1229.4 (8.9)** | **1218.3 (12.7)** | 1213.3 (10.3) | 1207.1 (4.9) |
| | | 10000 | 1194.3 (14.7) | 1156.4 (14.5) | 991.5 (38.6) | **1198.1 (11.3)** | 1193.2 (15.6) | 1165.1 (13.2) | 1178.2 (10.0) | 1179.7 (20.7) | **1218.6 (15.4)** | 1155.0 (16.6) |

