# OpenReview forum: "Improving Generative Adversarial Imitation Learning with Non-expert Demonstrations"
_ICLR.cc/2019/Conference_

### Official Review · AnonReviewer3 · 2018-11-02
**too little contribution**

**Rating:** 4
**Confidence:** 5

**Review:**

The paper proposes a modification of GAIL (Ho & Ermon, 2016) to make use of non-expert data. The non-expert data is used by training a classifier to classify between roll-outs of the current policy, expert demonstrations and non-expert demonstrations. Similar to GAIL, the policy is iteratively updated using TRPO with a cost that is given by the log probability of predicting the policy. The use of non-expert data acts as regularization in order to learn better features similar to universum prescription (Zhang & LeCun, 2017).

The paper is well-written and very clear. The general problem setting is interesting, but I think it is of rather little significance, because I do not see many clear applications. The evaluation focuses on simulated robots, however gathering non-expert data on real robots would be very expensive, so the approach would not make a lot of sense here (even if we replace TRPO a more sample efficient rl method). The paper mentions the game of Go, but learning a policy on such large state spaces is not feasible without major modification and significant computational effort. However, the paper also mentions autonomous driving, which might be a more convincing application, because we can have a lot of demonstrations that we do not want to label as expert trajectories. I think the paper would profit a lot from having an experiment where the importance of making use of non-expert data becomes evident.

The approach seems sound, although I think that we can not expect much benefit from using the unlabelled data in the proposed way. By not making any assumptions on the non-expert data, they do not carry any information about the objective; the information that they carry about the system dynamics is not exploited for the RL update. Instead, the use of non-expert data is restricted to learning better features for discriminating between the agent and the expert. However, non-expert data is typically not cheaper than policy roll-outs and better features could also be learned by using more samples from the policy. The experiments also show only slight benefits, especially when comparing the final performance (instead of total returns) and accounting for the additional system interactions needed for generating the non-expert data. To make the comparison fairer, we could consider using a few more system interactions (variable K in the paper) per iteration for standard GAIL, so that the total number of function evaluations would match those of M-GAIL after a certain number of iterations. Especially if K is appropriately tuned, it is not clear whether we could still show an advantage of M-GAIL. It would also be interesting to show, whether we can benefit from using the policy roll-outs of previous iterations as non-expert data for the current iteration (in the traditional IL setting where no non-expert data is available a priori).

The main weakness of the paper is, that the novelty seems marginal. Instead of doing binary classification with cross-entropy loss, we're doing three-class classification with cross-entropy loss and use it for binary classification (by throwing away the auxiliary logit for predicting the non-expert class). Did I miss any other difference to GAIL? We can argue whether the policy objective is different (to me, H_\phi of M-GAIL corresponds to the discriminator of GAIL and the objective is exactly the same), however, even if we call it a minor modification, we would have very little novelty in the approach. As the paper does also not compensate for this with very good results or thorough theoretical analysis, I think that the contribution is too minor.

I do not see the point of section 4.4. and the related appendix A2. For all I understand, it proves that when using lambda=0 (standard GAIL, right?), the proof of Fu et al. (2018) for GAIL is valid (i.e. we learn a completely useless reward function that does not carry any additional information compared to the policy), and when using lambda!=0 we learn something different. I don't see the the purpose of this statement and I don't think that it needs to be proven. The paper argues, that for small lambda we can treat the discriminator logits as approximations of these (completely useless) reward functions--without providing any bound. As I do not see why this would be useful, I think the section should be removed.

Minor:
Typo: "[...]due to its dependent[sic] on the linearity of reward functions and good feature engineering"

---

> ### Author Response · Authors · 2018-11-23
> **Reply to reviewer 3's comment**
>
> Thank you for the comments. Our replies to the comments and questions of reviewer 3 are given below.
>
> 1. The general problem setting is interesting, but I think it is of rather little significance, because I do not see many clear applications.
>
> -We would like to argue that there are many applications, including robotics, where non-expert demonstrations are abundance despite them being expensive to obtain. For instance, kinesthetic teaching is often used to gather demonstrations from a human teacher in robotics. However, it is common that the teacher does not know physical constraints of the robot well initially and the teacher needs some time to be familiar with the robot. During this time, teacher’s demonstrations, which have a large margin of error and are sub-optimal, would be throwaway for traditional IL methods. The expert demonstrations are collected only when the teacher is well familiar with the robot and can perform the task with a minimal error (Small errors and noises in demonstrations can be coped with by probabilistic formulation of maximum entropy IRL and GAIL, as mentioned by reviewer 4.).
>
> While our experiments with simulated robots are not exactly the same as this example, we believe that they can serve as benchmarks to show the benefit of using non-expert demonstrations that are generally ignored. Indeed, evaluations on real-world applications are needed to firmly demonstrate the practical usefulness of this setting. However, such evaluations are beyond the scope of the paper which is to introduce the problem setting and propose a simple yet efficient approach to solve it.
>
> 2. I think the paper would profit a lot from having an experiment where the importance of making use of non-expert data becomes evident.
>
> - We agree, and we have included a new set of experiment with smaller numbers of samples available. The new results show that methods that use the non-expert dataset, including our method and the two new baselines, significantly outperform GAIL both in terms of average performance and final performance.
>
> 3. However, non-expert data is typically not cheaper than policy roll-outs.
>
> - While the complexity of collecting non-expert demonstrations and policy roll-outs are similar, non-expert demonstrations are more informative than roll-outs from a poor policy such as an initial policy. Moreover, collecting data by a non-expert human is more preferable since non-expert is more likely to execute safe actions while the agent may execute hazardous actions due to exploration noise.
>
> 4. The experiments also show only slight benefits, especially when comparing the final performance
>
> - We agree that the experiment setup seems unfair to GAIL since it uses less samples. We expect that the two new baselines make the evaluation of M-GAIL fairer. However, in terms of the total sample size, the agent collects samples of size 10 million in total while the non-expert dataset is only of size 0.1 million which is relatively small compared to 10 million.
>
> We also agree that final performance is important for evaluation. In the revision, we report final performance in Table 2 and Table 4. The results for the small sample setting in Table 2 show significant improvement when using non-expert demonstrations. For the large sample setting, there is no significant difference in the final performance in Table 4. This is as expected since GAIL eventually learns a good policy after it collects enough samples in this setting.
>
> 3. It would also be interesting to show, whether we can benefit from using the policy roll-outs of previous iterations as non-expert data for the current iteration
>
> - The suggested self-improving idea is very interesting, and we discuss this suggestion in the conclusion. However, we think that this approach can be highly unstable since the two classes of agent’s trajectories are changing in each iteration.
>
> 4. The main weakness of the paper is, that the novelty seems marginal.
>
> - While simple, the modification from binary classification to multiclass classification is very important to ensure that the agent learn the expert policy while utilizing non-expert dataset. Also, we would like to emphasize that one of our contribution is to demonstrate that universum learning is useful in IL and generative adversarial learning. To the best of our knowledge, this was never considered before despite successes of semi-supervised learning in IL and generative adversarial learning.
>
> 5. The point of section 4.4. and the related appendix A2.
>
> -The point of Section 4.4 is to show that the proposed method does not perform IRL and only approximates an IRL solution. We believe that this negative result will motivate developing deep IRL methods for the proposed problem setting. However, we agree that this result does not play an important role in the paper. For better clarity, we move Section 4.4 to appendix and discuss the relation only in the conclusion.

---

> > ### Comment · AnonReviewer3 · 2018-12-05
> > **I still think that my raised issues are valid.**
> >
> > Addressing 1)
> > The submission assumes a huge amount of non-expert data (2-3 orders of magnitude more than expert data). I think that it is not realistic to assume that such amounts of data are collected while familiarizing with kinesthetic teaching. Furthermore, in practice, we often need to adjust data collection while familiarizing with the task, and collecting/post-processing data can be laborious as well. Hence, I still don't think that the approach is relevant for robotics.
> >
> > Addressing 2)
> > I am mainly missing an experiment that shows a useful application of the approach. It is not very surprising that using a huge amount of non-expert data is better than not using it. However, can you show the benefit of the approach on a problem with a realistic split of expert and non-expert data?
> >
> > Addressing first 3)
> > I don't think that we can argue about expert data collection being safer than autonomous data collection, when using an approach that relies on a large amount of autonomous system interactions.
> >
> > Addressing both 4s)
> > I still think, that the theoretical contribution is very small and the practical benefits are marginal as well. It seems rather obvious that additionally using a large amount of close-to-expert data can improve the efficiency of GAIL and any IL/IRL approach in general. Hence, I do not think that demonstrating this is a significant contribution. The submission neither shows a convincing application for the problem setting, nor a convincing method to make use of the additional data. V-GAIL and U-GAIL are very naive baselines and still, they perform similar and sometimes even outperform M-GAIL. It is not enough to perform better than GAIL when using a massive amount of additional information, if you can not beat naive baselines that use the same amount of information. There are various other ways of using this data somehow that can undoubtedly benefit from it, e.g. initializing the GAIL policy with BC trained on all data, building a regularizer from  the non-expert data (and potentially decreasing its influence during optimization), fitting a distribution to the non-expert data to learn the discriminator using importance weighting, and many more. I would expect an ICLR/NIPS/ICML/AAAI paper on this topic to outperform such naive baselines by several orders of magnitude in terms of the required number of non-expert data.
> >
> > Addressing 5)
> > Thanks for removing this section.

---

### Official Review · AnonReviewer2 · 2018-11-03
**Intersting idea**

**Rating:** 7
**Confidence:** 3

**Review:**

Summary:
This paper is about adversarial imitation learning and how data from non experts can be used to improve representation learning of the discriminator function. They also change the training objective because minmax does not lead to an optimal policy in the proposed setting. The data from non experts is used a separate class (so the discriminator learns to discriminate between expert, agent and non-expert policy).

Clarity: well written, with an intro to both relative fields - reinforcement learning and imitation learning.

Comments:
Overall, quite a neat idea - the information from non expert policy can help with representation learning, and the authors show that it is indeed the case via a number of experiments
At the same time, i wonder if the third class is required. It seems that gail in the experiments eventually reaches the same performance (figure 1) by looking at more agents trajectories. why can't non expert trajectories be considered as an initial set of agent trajectories? If multiclass is indeed required, it would be nice to see a comparison of what happens when non experts trajectories were just considered agents
Another thing i am surprised about is the leel of variance of GAIL in Figure 1. In table 1 we see that standard errors between the new method and gail are comparable, where does such a huge diff in var come from in Figure 1?
Also the sensitivity of the algorithm to lambda - how would one go setting it? In the experiments it seems that authors just try two different values (0.1 and 0.5) but i assume this really should be a hyperparameter search for this. Is lambda dependent on the number of expert and non expert demonstrations that are available?

---

> ### Author Response · Authors · 2018-11-23
> **Reply to reviewer 2's comment**
>
> Thank you for the comments. Our replies to comments and questions of reviewer 2 are given below.
>
> 1. Why can't non-expert trajectories be considered as an initial set of agent trajectories? If multiclass is indeed required, it would be nice to see a comparison of what happens when non experts trajectories were just considered agents.
>
> - It is possible to include non-expert trajectories into agent trajectories and perform GAIL with binary classification. However, as mentioned in the common reply, this approach learns a mixture policy and does not perform IL. It also does not provide much improvement over GAIL in our experiments.
>
> An alternative approach is to use non-expert trajectories for off-policy update in policy learning. However, this is not applicable since we do not know the non-expert policy, so we cannot compute importance weight to correct the bias.
>
> 2. In table 1 we see that standard errors between the new method and gail are comparable, where does such a huge diff in var come from in Figure 1?
>
> - Previously in Table 1, we treated test return in each iteration of each trail as an individual sample. This makes the standard error very small since it is computed from a very large number of samples. We correct this misleading standard error in the revision and treat a total return of each trial as one sample. However, using only 5 samples (from 5 trials) we cannot evaluate statistical significance well. Instead, we consider two best methods in terms of the mean value in the new tables.
>
> 3.  the sensitivity of the algorithm to lambda. how would one go setting it? In the experiments it seems that authors just try two different values (0.1 and 0.5) but i assume this really should be a hyperparameter search for this. Is lambda dependent on the number of expert and non expert demonstrations that are available?
>
> - An optimal \lambda should depend on number of expert and non-expert demonstrations, similarly to a class prior in standard classification. However, it is not exactly a class prior since G_{\phi} also contains \lambda. Currently, we treat \lambda as a hyper-parameter specified by the user. In the revision, we include \lambda=0.01 in the experiments to better investigate the sensitivity. Developing a more principal approach (e.g., a computationally efficient online cross-validation procedure) to determine the optimal value of \lambda is left for future work.

---

### Official Review · AnonReviewer4 · 2018-11-08
**Formulation seems unnecessary compared to existing imitation learning frameworks**

**Rating:** 5
**Confidence:** 4

**Review:**

This paper proposes M-GAIL, which performs imitation learning from expert as well as sub-optimal demonstrations. This work builds off of GAIL (Ho et. al 2016) but modifies the discriminator with an additional class corresponding to sub-optimal demonstrations. The authors show empirically that including these sub-optimal demonstrations into the training process leads to faster and improved learning.

While allowing the use of sub-optimal demonstrations is important and could have useful benefits in practice, I find the new algorithmic formulation unnecessary when compared to previous works. One of the theoretical benefits of MaxCausalEnt IRL (and by extension GAIL) is that because the model is probabilistic, not all demonstrations need to be reward-maximizing. The definition of optimality is relaxed from meaning exact reward maximization to coming from some optimal distribution over trajectories.

Correct me if I am wrong, but in the case of this work, the proposed algorithm seems equivalent to adding the "sub-optimal" demonstrations to the expert demonstrations, but down-weighting each sub-optimal demonstration be a factor of \lambda. Thus, I don't believe we need to introduce an entirely new algorithm, with the concept of a 3rd class, to solve this problem. The existing MaxEnt frameworks seem to handle the notion of sub-optimality proposed in this paper just as well, interpreting the "sub-optimal" demonstrations as low-probability expert demonstrations. This feels cleaner and more intuitive than the current proposed explanation as maximizing a mixture of two reward functions (in the IRL view) or minimizing occupancy measure divergence over 3 distributions (in the IL view). If this equivalence is true, I would encourage the authors to include this discussion in the main paper, and if not, discuss the differences and possibly compare against this simple strategy as a baseline.

I find the empirical results quite interesting, even though the gains seem small. Including sub-optimal demonstrations to learn from could be a nice trick to improve the learning of GAIL-like algorithms, which is nice to know. I am curious if the authors tried annealing the \lambda term from 1.0 to 0.0, as lambda=1.0 is likely easier to learn from, but lambda=0.0 would have better asymptotic performance.

---

> ### Author Response · Authors · 2018-11-23
> **Reply to reviewer 4's comment**
>
> Thank you for the comments. Our replies to comments and questions of reviewer 4 are given below.
>
> 1. The proposed algorithm seems equivalent to adding the "sub-optimal" demonstrations to the expert demonstrations, but down-weighting each sub-optimal demonstration be a factor of \lambda. […] The existing MaxEnt frameworks seem to handle the notion of sub-optimality proposed in this paper just as well, interpreting the "sub-optimal" demonstrations as low-probability expert demonstrations. […] If this equivalence is true, I would encourage the authors to include this discussion in the main paper, and if not, discuss the differences and possibly compare against this simple strategy as a baseline.
>
> - A direct application of maximum entropy IRL is not appropriate in our setting since we assume that we have a much larger number of non-expert demonstrations. By mixing the two datasets, maximum entropy IRL treats expert demonstrations as low probability ones instead. To avoid this, it is necessary to down-weight non-expert demonstrations by a weight \lambda. We call this method U-GAIL in the revision.
>
> In U-GAIL, the agent learns a mixture policy where the mixture coefficient is controlled by \lambda. However, by using binary classification, or equivalently one reward function in the IRL view, the agent cannot distinguish between learning signals from expert and that from non-expert. As a result, we cannot choose to learn only the expert policy from an already learned discriminator. Using a third class avoids this issue since learning signals from expert and non-expert are separately given by different discriminators. For these reasons, a third class is necessary to perform IL using non-expert demonstrations, and our method is not equivalent to adding down-weighted non-expert demonstrations to expert demonstrations and then performing GAIL.
>
> While U-GAIL does not perform IL, our experiment in Section 6 shows that learning a mixture policy can be beneficial when non-expert is not too poor. To make use of this benefit, Section 4.4 considers a simple extension of M-GAIL that learns also a mixture policy but allows using different weights for the discriminator and policy learning steps.
>
> 2. I am curious if the authors tried annealing the \lambda term from 1.0 to 0.0, as lambda=1.0 is likely easier to learn from, but lambda=0.0 would have better asymptotic performance.
>
> - We have previously tried annealing \lambda for our method and we found that it is highly unstable. This is likely because changing \lambda also changes the optimal discriminator which is also coupling with changes in the agent’s trajectory distribution. Moreover, it is difficult to use the same annealing schedule for all tasks. For this reason, we keep \lambda fix which already gives good empirical performance.

---

> > ### Comment · AnonReviewer4 · 2018-11-24
> > **Reply to rebuttal**
> >
> > Thank you for responding to my concerns. The difference between M-GAIL, GAIL, and GAIL with sub-optimal demonstrations (i.e. U-GAIL) seem very minor theoretically, and there does not appear to be an appreciable performance difference between GAIL with sub-optimal demonstrations and M-GAIL experimentally. Thus, I do not believe I need to revise my score, as the difference between this work and previous work for this version of the paper is too minor.

---

### Official Review · AnonReviewer1 · 2018-11-10
**Interesting idea but experimental comparison may not be fair, also sensitivity to lambda parameter is unclear**

**Rating:** 5
**Confidence:** 3

**Review:**

Description:

This paper presents a variant of imitation-based reinforcement learning, when in addition to example trajectories of an expert , example trajectories of non-experts are available.

In brief, the method is a variant of the GAIL method for adversarial training. In GAIL, the policy is optimized to minimize the ability to discriminate (classify) two classes: trajectories of the expert vs. trajectories from the policy, where the discrimination ability is measured by a neural network discriminator function optimized for maximal discriminative ability. In the proposed  method "M-GAIL", the idea is that the discriminator is forced to also discriminate a third class of non-expert demonstrations, but policy optimization is done ignoring the non-expert demonstrations and classifying only the usual two classes with the discriminator.

The M-GAIL method is compared to GAIL on four control tasks with different amounts of simulated non-expert demonstrations available, and  it outperforms GAIL if the simulated non-expert demonstrations are chosen to be relatively good ones (simulated from a policy having 70% of expert performance).


Evaluation:

The method is described relatively well and the idea of incorporating nonexpert demonstrations seems sound.

It is not clear to me if the experiment is fair, since M-GAIL learns from more data than GAIL which learns from the expert demonstrations only. It is unclear to me why the experiments did not attempt to supply the nonexpert demonstrations to GAIL too - one could e.g. have naively pooled the nonexpert demonstration into one of GAIL's binary classes, "expert" or "policy". This is especially concerning since bett

The method also requires the additional parameter lambda which affects performance in the experiments - it is not clear how to set it in practice, does e.g. cross-validation etc. need to be used? It would be useful to know more about sensitivity to lambda, experiments only consider two values.


Additional comments:

In the methodological derivation, it was unclear to me why only one parameter lambda is used to control class balance, why not two parameters controlling prevalence of the expert class, policy class, and nonexpert class?

In proposition 1 the fact that the bias vanishes when lambda=0 seems trivial because eq. 9 reduces to the first term on the right hand side.

In eq. 5 it's not quite right to call the right-hand side a loglikelihood. Loglikelihoods should be sums over observations of each class (thus emphasizing classes with more data) whereas here each term is an expectation - or do you assume the number of samples corresponding to each expectation term is equal?

Clarify the notation d_phi when you introduce it near eq. 3.

---

> ### Comment · AnonReviewer1 · 2018-11-10
> **Correction to broken sentence**
>
> The sentence "This is especially concerning since bett" was missing some words, it should have read as follows: "This is especially concerning since best performance in M-GAIL depends on quality of the non-expert demonstrations, hence GAIL could also benefit from them under some quality settings."

---

> > ### Author Response · Authors · 2018-11-23
> > **Reply to reviewer 1's comment**
> >
> > Thank you for the comments. Our replies to the comments and questions of reviewer 1 are given below.
> >
> > 1. It is unclear to me why the experiments did not attempt to supply the nonexpert demonstrations to GAIL too - one could e.g. have naively pooled the nonexpert demonstration into one of GAIL's binary classes, "expert" or "policy".
> > - We have considered the suggested approaches before. However, we did not pursue them since in order to learn the expert policy they require annealing the weight parameter and eventually ignoring non-expert data. However, we agree that these approaches should have been discussed and evaluated in the paper. In the revision, we provide detailed discussions and evaluations of these approaches.
> >
> > 2. The method also requires the additional parameter lambda which affects performance in the experiments - it is not clear how to set it in practice, does e.g. cross-validation etc. need to be used? It would be useful to know more about sensitivity to lambda, experiments only consider two values.
> > - Hyper-parameter selection procedure such as cross-validation (CV) may be used to choose \lambda based on classification accuracy. However, the agent’s trajectory distribution is always changing and the optimal \lambda should be different between iterations. Since performing CV in every iterations is costly, we treat \lambda as a hyper-parameter specified by the user. To better investigate the sensitivity, we include \lambda=0.01 in the experiments.
> >
> > 3. In the methodological derivation, it was unclear to me why only one parameter lambda is used to control class balance, why not two parameters controlling prevalence of the expert class, policy class, and nonexpert class?
> > - It is possible to use additional weights to balance the three classes since they only change weights of each occupancy measure in the JS divergence. We decide to have only the weight parameter for the non-expert class since the standard GAIL assumes the same class balance between expert and agent classes.
> >
> > 4. In proposition 1 the fact that the bias vanishes when lambda=0 seems trivial because eq. 9 reduces to the first term on the right hand side.
> > - Proposition 1 is a trivial result based on that of Fu et al. Our intentions are to show that M-GAIL only finds an approximate IRL solution and that further study is needed to perform IRL in our setting. To make this clearer, we only briefly mention the relation to IRL in the conclusion and move detailed discussions to appendix.
> >
> > 5. In eq. 5 it's not quite right to call the right-hand side a loglikelihood. Loglikelihoods should be sums over observations of each class (thus emphasizing classes with more data) whereas here each term is an expectation - or do you assume the number of samples corresponding to each expectation term is equal?
> > - Thank you for pointing out this mistake. We do not specifically assume equal sample sizes. We correct the terminology and use the term objective instead of log-likelihood in the revision.

---

### Author Response · Authors · 2018-11-23
**Changes made in the revision**

Dear reviewers,

We thank the reviewers for insightful comments and suggestions. Our replies to reviewers’ comments and questions will be posted separately. In this comment, we explain changes in the revision that should address concerns of the reviewers.

---Two GAIL extensions in Section 5 that mix the non-expert dataset to the expert or agent datasets.---
Common major concerns of the reviewers are the importance of using the third class and the lack of comparison against GAIL extensions that perform binary classification with non-expert data. To clarify these, we add a new section (Section 5) titled “Discussion on Binary Classification with Non-expert Data” that discusses two extensions of GAIL. In the first extension which we call U-GAIL, the non-expert dataset is mixed with the expert dataset and is down-weighted by \lambda. This method is equivalent to the method suggested by reviewer 1 and 4. An important issue of this method is that it learns a mixture of expert and non-expert policies. Thus, it does not learn only the expert policy which is the goal of IL. Our experiments show that this method works well when non-expert policy is not too different from the expert policy but works poorly otherwise.

For the second extension which we call V-GAIL, the non-expert dataset is mixed with the agent (current policy) dataset and is down-weighted by \lambda. This method is equivalent to the method suggested by reviewer 1 and 2. Similarly to U-GAIL, this method does not perform IL since it learns a policy mixture of expert and non-expert policies. Moreover, it requires an additional constraint to ensure non-negativity of the resulting occupancy measure. Our experiments show that this method does not provide much improvement over GAIL.

The issue of these two extensions emphases the novelty and importance of performing multiclass classification with non-expert demonstrations in our method.

We also evaluate a simple extension of our method where we learn a policy mixture, similarly to U-GAIL. We call this method mixture M-GAIL (MM-GAIL) and discuss it in Section 4.4. Unlike U-GAIL, we can specify a mixture coefficient (\omega) during policy learning that can be different from \lambda which is used during discriminator learning.

---New experimental setting in Section 6 with smaller numbers of expert demonstrations.---
Another common concern of the reviewers is that non-expert dataset seems to provide only marginal improvement. To better demonstrate the usefulness of our method, we perform additional experiments with smaller numbers of expert demonstrations (N=100, 300, 500) and agent’s transition samples in each iteration (K=1000). The new results show that non-expert demonstrations significantly improve performance when compared to GAIL. This additional experiment can be found in the first half of Section 6. The previous experiment with N=1000, 10000 and K=2000 can be found in the second half of Section 6.

We also evaluate performance with \lambda=0.01 to investigate the sensitivity against \lambda of each method. The results in Figures 4 to 11 in the appendix show that M-GAIL is less sensitive to \lambda when compare to U-GAIL. This is likely because \lambda directly affects both the discriminator and the optimal policy (i.e., the mixture policy) in U-GAIL, while it only affects the discriminator in M-GAIL.

---

### Meta-Review · Area_Chair1 · 2018-12-15

**Confidence:** 4
**Recommendation:** Reject

**Metareview:**

This paper proposes a variant of GAIL that can learn from both expert and non-expert demonstrations. The paper is generally well-written, and the general topic is of interest to the ICLR community. Further, the empirical comparisons provide some interesting insights. However, the reviewers are concerned that the conceptual contribution is quite small, and that the relatively small conceptual contribution also does not lead to large empirical gains. As such, the paper does not meet the bar for publication at ICLR.